# Strategic A/B testing via Maximum Probability-driven Two-armed Bandit

**Yu Zhang** [* 1]  **Shanshan Zhao** [* 2]  **Bokui Wan** [3]  **Jinjuan Wang** [4]  **Xiaodong Yan** [5]

## Abstract

Detecting a minor average treatment effect is a major challenge in large-scale applications, where even minimal improvements can have a significant economic impact. Traditional methods, reliant on normal distribution-based or expanded statistics, often fail to identify such minor effects because of their inability to handle small discrepancies with sufficient sensitivity. This work leverages a counterfactual outcome framework and proposes a maximum probability-driven two-armed bandit (TAB) process by weighting the mean volatility statistic, which controls Type I error. The implementation of permutation methods further enhances the robustness and efficacy. The established strategic central limit theorem (SCLT) demonstrates that our approach yields a more concentrated distribution under the null hypothesis and a less concentrated one under the alternative hypothesis, greatly improving statistical power. The experimental results indicate a significant improvement in the A/B testing, highlighting the potential to reduce experimental costs while maintaining high statistical power.

## 1. Introduction

**Background and Motivation.** A / B testing, which is ubiquitous in tech companies, has become the gold standard for evaluating the discrepancy between new and existing strategies (Kohavi et al., 2009; 2013), providing crucial information for decision-making. Having efficient and reliable methods for A/B testing is critical in accelerating the pace of strategic improvement (Pearl, 2009; Kong et al., 2022).

[*]Equal contribution [1]Zhongtai Securities Institute for Financial Studies, Shandong University, Jinan, China [2]School of Mathematics, Shandong University, Jinan, China [3]Didi Chuxing, Beijing, China [4]School of Mathematics and Statistics, Beijing Institute of Technology, Beijing, China [5]School of Mathematics and Statistics, Xi'an Jiaotong University, Xian, China. Correspondence to: Xiaodong Yan <yanxiaodong@xjtu.edu.cn>.

*Proceedings of the 42$^{nd}$ International Conference on Machine Learning*, Vancouver, Canada. PMLR 267, 2025. Copyright 2025 by the author(s).

In practice, to facilitate a comparison between two strategies (e.g., two distinct ad configurations), users of the platform are randomly divided into two groups, with each group exposed to one of the test strategies. Subsequently, the interactions of the subjects with the site or app are recorded. After the experiment is completed, the performance of the two groups will be compared in several key metrics, such as the estimated mean difference in click counts or the volume of gross merchandise (GMV), to determine which strategy is preferred (Hohnhold et al., 2015; Xu & Chen, 2016).

**Goal.** Let $A = 1$ and 0 denote that the subjects are assigned to the treatment group (new strategy) and the control group (existing strategy), with a potential outcome $Y^{(1)}$ and $Y^{(0)}$, respectively. The interest that whether the treatment group significantly outperforms the control group can be formalized by

$$\mathcal{H}_0 : \mu \le 0 \quad \mathcal{H}_1 : \mu > 0, \tag{1}$$

where $\mu = \mathbb{E}(Y^{(1)} - Y^{(0)})$ denotes the average treatment effect. This test is widely adopted in the decision-making market, such as one of the world's leading ride-sharing companies, to quantify the "`Causal Discrepancy`" between the A / B strategies. A new strategy can be accepted if it demonstrates statistically significant performance improvements over the existing one (Yao et al., 2021).

**Challenge.** Normal distribution-based or expanded test statistics are unable to statistically identify minor improvements of the average treatment effect $\mu$.

- **Minor average treatment effect (ATE)**. Practically, the marginal returns of the strategies become increasingly indistinguishable as the business expands, and conventional A/B testing methods are unable to statistically identify minor improvements (Kohavi et al., 2013). However, due to economies of scale, any benefit from improved sensitivity will be magnified, as even minor differences detected in key metrics can have a significant impact on total revenue (Deng et al., 2013). To illustrate, a strategy that increases revenue by \$0.02 per user could generate millions of dollars in total revenue for fifty million users. Alternatively, enhanced sensitivity facilitates the conduct of experimentation in smaller user groups or over shorter time periods

while maintaining equivalent statistical power, thereby reducing experimental costs.

- **Exchangeable integration**. Existing approaches in this field can generally include G-computation that constructs regression models between outcomes and covariates (Keil et al., 2014; Wang et al., 2017), propensity score (PS) based methods that employ the PS to adjust the response (Williamson et al., 2014; Austin & Stuart, 2017), and doubly robust methods that combine the principles of G-computation and PS (Zhang et al., 2012; Chernozhukov et al., 2018). But the aforementioned methods are constrained by utilizing normal distribution-based statistics for the purpose of hypothesis testing minor ATE.

**Contributions**. This work focuses on minor ATE detection based on the counterfactual outcome framework and breaks the exchangeable structure of original data by a novel two-armed bandit model. Our contributions are summarized as follows:

- **Methodologically**, this work proposes a new maximum probability-driven TAB process by weighting the mean volatility statistic (Chen et al., 2022; 2023) for a more powerful A/B testing. By modifying the weights assigned to the mean and volatility terms in the statistic, it is available to control the type I error. And based on the the permutation and $p$-value combination framework originally developed in Wang et al. (2025), we achieve enhanced robustness while improving statistical power.

- **Theoretically**, we develop a new strategic central limit theorem (SCLT) under the optimal ranking policy of the data in a larger probability space. The proposed weighted mean-volatility statistic demonstrates greater concentration under the null hypothesis and less concentration under the alternative than the classic central limit theorem (CLT), thereby enhancing the statistical power.

## 2. Preliminaries and Related Work

### 2.1. A/B testing and variance reduction

**A/B testing without confounders**. The purpose of the A/B testing is to ascertain whether the strategy of treatment group exhibits significant superiority in comparison to that of the control group. In this paper, we focus on statistical inference for the average treatment effect (ATE), as illustrated in (1). Let $A$ denote a binary treatment indicator, where $A = 1$ means that subjects are assigned to the treatment group, and $A = 0$ indicates assignment to the control group. $Y$ is employed to represent some interesting outcome metrics

observed from the subjects. Denote the number of samples with $A = 1$ and $A = 0$ by $n^{(1)}$ and $n^{(0)}$, respectively. Following the Rubin Causal Model (RCM) (Holland, 1986; Imbens & Rubin, 2010), let $Y^{(1)}$ and $Y^{(0)}$ be the corresponding potential outcome when the subject is assigned with the treatment or not. When there is no confounder, the common difference in the mean estimator (DIM) for ATE can be constructed as:

$$\text{DIM} = \sum_{i:A_i=1} Y_i/n^{(1)} - \sum_{i:A_i=0} Y_i/n^{(0)},$$

with its variance var(DIM)$= \widehat{\sigma}_1^2/n^{(1)} + \widehat{\sigma}_0^2/n^{(0)}$, where $\widehat{\sigma}_1^2$ and $\widehat{\sigma}_0^2$ are the sample variances in the treatment and control groups, respectively.

**A/B testing with confounders**. However, DIM is no longer an unbiased estimate of ATE in instances where there exists confounding variables. Consequently, some state-of-the-art approaches have been proposed to overcome the impact of confounders. Existing approaches in this field can generally be classified into three main groups:

1. **G-computation**. G-computation, also known as the parametric g-formula or g-standardization, estimates the counterfactual outcomes by constructing a regression model (linear or non-linear) between the outcome metrics and the covariates (Robins, 1986; Snowden et al., 2011; Vansteelandt & Keiding, 2011; Keil et al., 2014; Wang et al., 2017).

2. **Propensity score-based methods**. These methods generate a pseudo population in which $A$ is independent of covariates, allowing the estimation of marginal structural model parameters and ATE (Hirano et al., 2003; Brookhart et al., 2006; Gayat et al., 2010; Williamson et al., 2014; Austin & Stuart, 2017).

3. **Doubly Robust methods**. These methods combine G-computation techniques and propensity score-based methods to achieve consistency under more relaxed conditions and to obtain a lower estimation variance (Bang & Robins, 2005; Tan, 2010; Funk et al., 2011; Zhang et al., 2012; Chernozhukov et al., 2018).

**Variance reduction.** In randomized controlled trials (RCTs), where no confounders are present, elevated variance and slow convergence speed result in a decreased efficacy of DIM-based hypothesis testing. One technique for reducing variance involves incorporating relevant data from the pre-experiment period as covariates to reduce the variability of the outcome metrics. CUPED (Deng et al., 2013) employs the linear correlation between the pre-treatment data and the outcome metrics to reduce the variance of the experimental and control groups, constructing unbiased estimators with lower variance. However, its effectiveness is

limited by the linear correlation assumption between the covariates and the outcome metrics. Consequently, to identify nonlinear relationships between covariates and outcome metrics, several machine learning (ML)-based estimators have been developed (Wu & Gagnon-Bartsch, 2018; Guo et al., 2021; Jin & Ba, 2023). CUPAC (Tang et al., 2020) and MLRATE (Guo et al., 2021) are two of the most advanced estimators in this category, both based on regression adjustment methods to capture more complex regression relationships.

**Note**: However, the aforementioned methods construct the test statistic under the data exchangeability and use the IID-based central limit theorem to study their asymptotic distributions under null and alternative hypothesis. The efficacy of normal distribution-based or expanded statistics for the purpose of testing the minor ATE is limited.

### 2.2. Two–armed bandit framework

Based on the framework of the strategic central limit theorem (Chen et al., 2022; 2023), data exchangeability can be broken to use data dependence to attain some objective, which aims at improving the testing power for the minor ATE in this work.

The two-armed bandit is a classic and widely studied model in reinforcement learning, which consists of three elements: the agent, the action space, and the reward space. Let $\vartheta_i \in \{0, 1\}$ represent the two arms of the bandit, where $i$ denotes the time step. At each step, the agent selects an arm $\vartheta_i$ and obtains an independent reward $R_i^{(\vartheta_i)}$. The decisions made by the agent at each time step are driven by the sequence of responses obtained so far and the goal of the two-armed bandit is to find an optimal policy $\theta_n^* = (\vartheta_1^*, \ldots, \vartheta_n^*)$ that maximizes the expected cumulative reward.

Drawing inspiration from the efficacy of the two-armed bandit problem in utilizing sequential data to maximize the given objective, Chen et al. (2022; 2023) proposed the implementation of a novel TAB process to construct a test statistic that maximizes the probability of the tail. This approach breaks the exchangeable structure and constructs a novel asymptotic distribution. To better illustrate, consider an oracle scenario in which the counterfactual outcomes of each subject $Y_i^{(1)}$ and $Y_i^{(0)}$ can be observed simultaneously. When arm $\vartheta_i = 0$ is selected, the reward $R_i^{(0)} = Y_i^{(0)} - Y_i^{(1)}$ is observed. In contrast, when arm $\vartheta_i = 1$ is chosen, reward $R_i^{(1)} = Y_i^{(1)} - Y_i^{(0)}$ is obtained. The following statistic can be derived under the policy $\theta_n = (\vartheta_1, \ldots, \vartheta_n)$:

$$T_n(\theta_n) = \underbrace{\frac{1}{n} \sum_{i=1}^{n} \bar{R}_n^{(\vartheta_i)}}_{\text{Mean}} + \underbrace{\frac{1}{\sqrt{n}} \sum_{i=1}^{n} \frac{R_i^{(\vartheta_i)}}{\widehat{\sigma}}}_{\text{Volatility}}, \qquad (2)$$

where $\widehat{\sigma}^2 = \sum_{i=1}^{n}(R_i^{(1)} - \bar{R}_n^{(1)})^2/(n-1)$ and $\bar{R}_n^{(1)} = \sum_{i=1}^{n} R_i^{(1)}/n = -\bar{R}_n^{(0)}$. This statistic is similar to a combination of the classic causal discrepancy index and traditional hypothesis testing statistics, providing a comprehensive perspective for performance evaluation.

From the expression of $T_n(\theta_n)$, it can be seen that if the baseline policy $\theta_n^b$ is employed: for any $n \geq 1$, $\mathbb{P}(\theta_n^b = \{1, 1, \ldots, 1\}) = \mathbb{P}(\theta_n^b = \{0, 0, \ldots, 0\}) = 0.5$, then the second term of $T_n(\theta_n)$ will simplify to the $z$-test:

$$\frac{\sum_{i=1}^{n} R_i^{(1)}}{\sqrt{n}\widehat{\sigma}} \quad \text{or} \quad \frac{\sum_{i=1}^{n} R_i^{(0)}}{\sqrt{n}\widehat{\sigma}}. \qquad (3)$$

Both test statistics converge asymptotically to a normal distribution, only with a distinct expectation. However, the information of $Y_i^{(1)} - Y_i^{(0)}$ for each $i$, including its sign and magnitude, is not fully utilized in the statistic $z$. The objective of the policy $\theta_n$ is to reconstruct the two statistics in (3) to make full use of the data information. The different policies correspond to different distributions of $T_n(\theta_n)$, as shown in Figure 1 (bottom). Therefore, the optimal policy $\theta_n^*$ for maximizing tail probability (Chen et al., 2022; 2023) is given by Lemma 2.1.

**Lemma 2.1.** *For any $n \geq 1$, we can construct the strategy $\theta_n^* = \{\vartheta_1^*, \ldots, \vartheta_n^*\}$ as follows: for $i = 1$, $\mathbb{P}(\vartheta_1^* = 1) = \mathbb{P}(\vartheta_1^* = 0) = 0.5$, and for $i \geq 2$,*

$$\vartheta_i^* = \begin{cases} 1, & \text{if } T_{i-1}(\theta_{i-1}^*) \geq 0, \\ 0, & \text{else.} \end{cases} \qquad (4)$$

*Under the guidance of the optimal policy $\theta_n^*$, $T_n(\theta_n^*)$ has the following property:* $\lim_{n \to \infty} \mathbb{P}(|T_n(\theta_n^*)| > z_{1-\alpha/2}|\mathcal{H}_1) = \lim_{n \to \infty} \sup_{\theta_n \in \Theta} \mathbb{P}(|T_n(\theta_n)| > z_{1-\alpha/2}|\mathcal{H}_1)$, *where $z_\alpha$ denotes the $\alpha$th quantile of a standard normal distribution and $\mathbb{P}(\cdot|\mathcal{H}_1)$ denotes the conditional probability given that the alternative hypothesis $\mathcal{H}_1$ is true.*

*Remark* 2.2. Lemma 2.1 shows that $\theta_n^*$ has the highest probability of rejecting the null hypothesis when the alternative hypothesis is true, minimizing the probability of a Type II error. The corresponding rejection region is the brown shaded area in Figure 1 (top).

## 3. Method

### 3.1. Weighted two-armed bandit (WTAB) process

**Oracle test via WTAB statistic**. Equation (2) provides an expression for the statistic $T_n(\theta_n)$, where the term $\bar{R}_n^{(\vartheta_i)}/n$ and the term $R_i^{(\vartheta_i)}/(\sqrt{n}\widehat{\sigma})$ are assigned equal weights. However, a more thorough study of $T_n(\theta_n)$ revealed that employing distinct weights for these two terms could potentially improve statistical power, as illustrated in Figure 2(a).

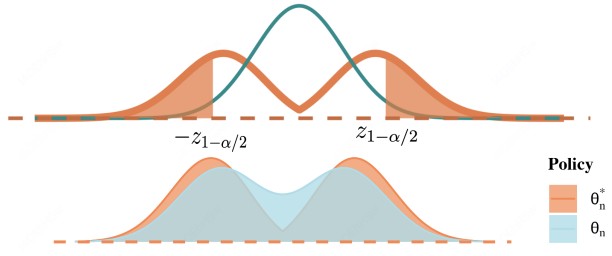

Figure 1: The power under the optimal strategy $\theta_n^*$ (brown shadow) (top) and the probability density plot of the test statistic $T_n(\theta_n)$ under different strategies (bottom).

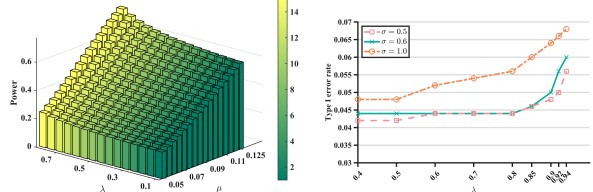

(a) Statistical power across different $\lambda$ and $\mu$.

(b) The empirical type I error rate across different $\lambda$ and $\sigma$, fixed $n = 20000$.

Figure 2: Plot of statistical power and empirical type I error rate as $\lambda$ varies. It shows that test efficacy follows a same trend to $\lambda$, with a concurrent risk of inflation in the type I error rate.

Specifically, rewrite $T_n(\theta_n)$ as:

$$T_{n,\lambda}(\theta_n) = \frac{1}{n}\sum_{i=1}^{n}\lambda\bar{R}_n^{(\vartheta_i)} + \frac{1}{\sqrt{n}}\sum_{i=1}^{n}(1-\lambda)\frac{R_i^{(\vartheta_i)}}{\hat{\sigma}}.$$

where $\lambda \in (0,1)$. Equivalently, to obtain a uniform form for the limiting distribution, $T_{n,\lambda}(\theta_n)$ can be rewritten as:

$$T_{n,\lambda}(\theta_n) = \frac{1}{n}\sum_{i=1}^{n}\frac{\lambda}{1-\lambda}\bar{R}_n^{(\vartheta_i)} + \frac{1}{\sqrt{n}}\sum_{i=1}^{n}\frac{R_i^{(\vartheta_i)}}{\hat{\sigma}}. \quad (5)$$

Intuitively, a change in the form of the statistic may result in a change in the optimal policy $\theta_n^*$. Surprisingly, the optimal policy $\theta_n^*$ corresponding to the statistic $T_{n,\lambda}(\theta_n)$ remains formally identical to that given in Equation (4). Theorem 4.1 and 4.2 show that this policy continues to be optimal in terms of maximizing statistical power.

The parameter $\lambda$ in $T_{n,\lambda}(\theta_n^*)$ serves to balance the trade-off between the Type I error rate and statistical power. As illustrated in Figures 2(a) and 3, increasing $\lambda$ generally leads to higher statistical power. However, according to Theorem 4.1, it is not reasonable to blindly pursue a larger $\lambda$ when constructing the test statistic. Specifically, for fixed values of $n$ and $\sigma$, an excessively large $\lambda$ can slow down the convergence of $T_{n,\lambda}(\theta_n^*)$, thereby affecting the efficacy of the test. This observation is further supported by Figure 2(b), which shows that a mismatch between $\lambda$ and the current values of $n$ and $\sigma$ can lead to inflated Type I error rates, ultimately resulting in an overestimation of the strategy's efficacy.

As demonstrated in Theorem 4.1, the presence of $\lambda$ has been shown to reduce the convergence rate of $T_{n,\lambda}(\theta_n^*)$ by $\lambda\sigma/((1-\lambda)\sqrt{n})$. Consequently, a threshold (typically 0.03) is selected to regulate the magnitude of $\lambda$, that is, to identify the largest that satisfies $\lambda\sigma/((1-\lambda)\sqrt{n}) \leq 0.03$. Alternatively, a data-driven approach can be employed to select an appropriate value of $\lambda$. Specifically, bootstrap methods (Hesterberg, 2011) can be combined with a type I error rate test to identify the optimal $\lambda$ that maximizes the statistical power while controlling the type I error.

In contrast to traditional hypothesis testing methods, which rely on the CLT and assume exchangeable data, the TAB framework introduces non-exchangeability. Specifically, the policy $\theta_n^*$ causes earlier samples to influence subsequent ones, making the construction of the test statistic $T_{n,\lambda}(\theta_n^*)$ dependent on the order of observations. By using SCLT, the test statistic no longer requires the normality assumption, enhancing statistical power while maintaining type I error control. Under the null hypothesis $\mu \leq 0$, the statistic $T_{n,\lambda}(\theta_n^*)$ exhibits a more centralized distribution around zero and satisfies $\mathbb{P}(|T_{n,\lambda}(\theta_n^*)| < z_{1-\alpha/2}) < \alpha$, when the alternative hypothesis $\mu > 0$ holds, it demonstrates a greater dispersion away from zero. The distinct distributions of $T_{n,\lambda}(\theta_n^*)$ allow us to effectively differentiate between them. Consequently, the two-tailed rejection region for one-sided hypothesis testing based on $T_{n,\lambda}(\theta_n^*)$ is defined as $|T_{n,\lambda}(\theta_n^*)| > z_{1-\alpha/2}$. Furthermore, Theorem 4.2 provides evidence to support the validity of $T_{n,\lambda}(\theta_n^*)$.

**Estimation of the counterfactual outcomes**. In contrast to oracle cases, a serious challenge under A/B testing is the issue of missing data, which means that only one outcome can be observed for each subject. A nuisance issue that requires resolution is the simultaneous acquisition of $Y_i^{(1)}$ and $Y_i^{(0)}$ for each subject. Given that the same subject will only be included in one of the control or treatment groups in an experiment, it is not feasible to observe their counterfactual results. Consequently, a pseudo population will be constructed for the purpose of estimating the missing data. According to the properties of conditional expectation and the assumptions of RCM, we can derive the following:

$$\begin{aligned}\mu &= \mathbb{E}(Y^{(1)} - Y^{(0)}) \\ &= \mathbb{E}\big[\mathbb{E}(Y^{(1)}|A=1,X)\big] - \mathbb{E}\big[\mathbb{E}(Y^{(0)}|A=0,X)\big] \\ &= \mathbb{E}\big[\mathbb{E}(Y|A=1,X)\big] - \mathbb{E}\big[\mathbb{E}(Y|A=0,X)\big]. \quad (6)\end{aligned}$$

Consequently, let $m_1(x)$ and $m_0(x)$ denote the outcome regression function, and $e(x)$ denote the propensity score function such that $m_1(x) = \mathbb{E}(Y|A=1,X=x)$, $m_0(x) =$

$\mathbb{E}(Y|A = 0, X = x)$ and $e(x) = \mathbb{P}(A = 1|X = x)$. Using the Doubly Robust (DR) estimator, Equation (6) can be rewritten as:

$$\mu = \sum_{a=0}^{1}(-1)^{a+1}\mathbb{E}\left[m_a(X) + \frac{\mathbb{I}(A = a)}{e_a(X)}\left(Y - m_a(X)\right)\right],$$

where $e_a(X) = e(X) \cdot \mathbb{I}(a = 1) + (1 - e(X)) \cdot \mathbb{I}(a = 0)$ and $\mathbb{I}(\cdot)$ denotes the indicator function. The use of a doubly robust estimator provides several advantages. Primarily, the estimator requires only that either the outcome regression models or the propensity score model be correctly specified to yield an unbiased estimate of ATE, a property known as double robustness. Secondly, in cases where both the outcome regression model and the propensity score model are correctly specified, the doubly robust estimator achieves lower variance compared to traditional propensity score-based methods (Bang & Robins, 2005; Chernozhukov et al., 2024). Once the estimates of the outcome regression models and the propensity score model have been obtained, they can be used to estimate the ATE. Drawing on the ideas from Wang et al. (2025), denote:

$$\widehat{\mu}_i = \sum_{a=0}^{1}(-1)^{a+1}\left[\widehat{m}_a(x_i) + \frac{\mathbb{I}(A_i = a)}{\widehat{e}_a(x_i)}\left(Y_i - \widehat{m}_a(x_i)\right)\right],$$

and $\widehat{\mu}_i^{(1)} = \widehat{\mu}_i = -\widehat{\mu}_i^{(0)}$, then $T_{n,\lambda}(\theta_n^*)$ can be constructed as:

$$T_{n,\lambda}(\theta_n^*) = \frac{1}{n}\sum_{i=1}^{n}\frac{\lambda}{1-\lambda}\bar{\mu}_n^{(\vartheta_i^*)} + \frac{1}{\sqrt{n}}\sum_{i=1}^{n}\frac{\widehat{\mu}_i^{(\vartheta_i^*)}}{\widehat{\sigma}}, \quad (7)$$

where $\widehat{\sigma}^2 = \sum_{i=1}^{n}(\widehat{\mu}_i^{(1)} - \bar{\mu}_n^{(1)})^2/(n - 1)$ and $\bar{\mu}_n^{(1)} = \sum_{i=1}^{n}\widehat{\mu}_i^{(1)}/n = -\bar{\mu}_n^{(0)}$. The use of sample variance as an estimator for true variance is supported by both theoretical and empirical evidence. Under the null hypothesis, after obtaining $\widehat{\mu}_i^{(1)}$, the $z$-test statistic is computed as $\sum_{i=1}^{n}\widehat{\mu}_i^{(1)}/(\sqrt{n}\widehat{\sigma})$, and its $p$-value distribution follows a uniform distribution $U(0, 1)$. This justifies the use of sample variance as an estimator of true variance, as detailed in Chernozhukov et al. (2018; 2024).

A key challenge lies in obtaining accurate estimates for functions $m_0(x)$, $m_1(x)$, and $e(x)$. Traditional methods such as CUPAC, which rely solely on linear regression, might fail to capture these intricate patterns. To overcome this limitation, advanced machine learning methods are introduced. Specifically, LightGBM (Ke et al., 2017), a state-of-the-art gradient boosting algorithm, is employed within the double machine learning (DML) framework (Chernozhukov et al., 2018). In practice, the observation dataset $\mathcal{D}$ is divided into $K$ equal subsets $D_k$. For each $D_k$, a LightGBM model is trained on the remaining data $\mathcal{D}/\mathcal{D}_k$ and applied to estimate counterfactual results of $\mathcal{D}_k$. This procedure is repeated for all subsets $\mathcal{D}_k$. Intuitively, the DML approach mitigates overfitting and reduces regularization biases by partitioning the dataset into multiple subsets. Each subset is used iteratively to estimate conditional relationships, ensuring robustness and improved predictive performance. (Chernozhukov et al., 2018; 2024). Additionally, XGBoost (Chen & Guestrin, 2016) is used to estimate $m_1(x)$, $m_0(x)$, and $e(x)$, exhibiting similar performance to LightGBM. To further improve efficacy, the ensemble learning technique stacking (Wolpert, 1992; Breiman, 1996) is applied, with LightGBM and XGBoost as the primary learners and linear regression serving as the meta-learner (Ting & Witten, 1999).

### 3.2. Permuted WTAB

As discussed previously, the $T_{n,\lambda}(\theta_n^*)$ statistic has demonstrated superior performance, but it is worth highlighting that it does not possess the property of sample order exchangeability, which implies that it is sensitive to the order in which the samples are presented, leading to the "$p$-value lottery" (Meinshausen et al., 2009). To address this issue, following the framework initially established in Wang et al. (2025), we employ sample permutation in WTAB testing coupled with Cauchy $p$-value combination. We perform multiple samples reorderings, repeatedly calculate the $p$-value of $T_{n,\lambda}(\theta_n^*)$, and aggregate these via meta-analysis to enhance the robustness of statistical inference. Specifically, first determine a number of permutations, denoted as $B$. Subsequently, the sequence $\{1, 2, \ldots, n\}$ is reordered into a new one by applying a mapping $\pi_b : \{1, 2, \ldots, n\} \to \{1, 2, \ldots, n\}$. For each element $i$ in the original sequence, its position in the reordered sequence is given by $\pi_b(i)$. For $b = 1, \ldots, B$, the mapping $\pi_b$ is applied to the counterfactual outcomes $\{\widehat{\mu}_i, i = 1, \ldots, n\}$, resulting in reordered samples $\{\widehat{\mu}_{\pi_b(i)}, i = 1, \ldots, n\}$. Finally, $T_{n,\lambda}^{(b)}(\theta_n^*)$ and its corresponding $p$-value $p_\lambda^{(b)}$ is calculated based on $\{\widehat{\mu}_{\pi_b(i)}, i = 1, \ldots, n\}$.

However, varying sample orderings can yield inconsistent $p_\lambda^{(b)}$ values, and the conclusions drawn from individual $p$-value may be unclear. To resolve this, we apply meta-analysis to synthesize an overall $p$-value, improving the reliability of the results derived from individual $p_\lambda^{(b)}$ (Walker et al., 2008; Lee, 2019). Several research studies are conducted on the subject of combining $p$-values, including Fisher's combination test (Fisher, 1970), quantile-based combination test (Meinshausen et al., 2009) and Cauchy combination test (Liu & Xie, 2020). Cauchy combination test is used for the combination of $p$-values in this paper, since it is straightforward and easy to implement and has also been shown to effectively aggregate multiple small effects, even when the $p$-values are dependent (Liu & Xie, 2020). The $p$-values are aggregated by Cauchy combination as follows:

**Algorithm 1** Permuted WTAB algorithm

---

**Input:** data $D = \{(X_i, Y_i, A_i), i = 1, \ldots, n\}$, threshold $\tau$, permutation times $B$.

**Output:** the aggregated $p$-value $p_a$.

1: $n_0 =$ number of samples in $D$.
2: Divide $D = \{(X_i, Y_i, A_i), i = 1, \ldots, n\}$ into $K$ disjoint subset $D_k$, each of equal size.
3: **while** $k \leq K$ **do**
4:     Estimate outcome regression functions $m_0(x)$, $m_1(x)$ and propensity score function $e(x)$ using LightGBM based on $D/D_k$, denoted by $\widehat{m}_0^{(k)}(x)$, $\widehat{m}_1^{(k)}(x)$ and $\widehat{e}^{(k)}(x)$;
5:     Substitute $D_k$ into the expression of $\widehat{\mu}_i$ to obtain the estimates of counterfactual outcomes.
6: **end while**
7: Estimate sample variance for the counterfactual outcomes as $\widehat{\sigma}^2 = \sum_{i=1}^{n} (\widehat{\mu}_i^{(1)} - \bar{\mu}^{(1)})/(n-1)$, where $\bar{\mu}^{(1)} = \sum_{i=1}^{n} \widehat{\mu}_i^{(1)}/n$.
8: Compute $\lambda = \tau\sqrt{n}/(\widehat{\sigma} + \tau\sqrt{n})$.
9: **while** $b \leq B$ **do**
10:     Get a set of reordered samples of $\{\widehat{\mu}_i, i = 1, \ldots, n\}$, denoted by $\{\widehat{\mu}_{\pi_b(i)}, i = 1, \ldots, n\}$;
11:     Calculate the test statistic $T_{n,\lambda}^{(b)}(\theta_n^*)$ based on $\{\widehat{\mu}_{\pi_b(i)}, i = 1, \ldots, n\}$ and its corresponding $p$-value $p_\lambda^{(b)} = 2\Phi\left(-\left|T_{n,\lambda}^{(b)}(\theta_n^*)\right|\right)$.
12: **end while**
13: Aggregate $\{p_\lambda^{(b)}, b = 1, \ldots, B\}$ by Equation (8).
14: Compute and output the aggregated $p$-value $p_a$ by Equation (9).
15: **Return** $p_a$.

---

$$C_B = \frac{1}{B} \sum_{b=1}^{B} \tan\left[(0.5 - p_\lambda^{(b)})\pi\right]. \quad (8)$$

The aggregated $p$-value is calculated as

$$p_a = 0.5 - \arctan(C_B)/\pi, \quad (9)$$

and subsequently employed to ascertain whether the null hypothesis should be rejected. In this paper, $B$ is set to 25, as it has been observed that increasing $B$ further does not substantially improve statistical power. The Algorithm 1 provides a comprehensive illustration of the proposed permuted WTAB algorithm (PWTAB).

## 4. Theoretical Properties

In this section, we first summarize the asymptotic properties of the oracle test statistic $T_{n,\lambda}(\theta_n^*)$ to demonstrate the practicality of the maximum probability-driven two-armed bandit framework.

**Theorem 4.1.** *Let* $\varphi \in C(\overline{\mathbb{R}})$*, the set of all continuous functions on* $\mathbb{R}$ *with finite limits at* $\pm\infty$*, be an even function and monotone on* $(0, \infty)$*. For any* $n \geq 1$*, construct the oracle test statistic* $T_{n,\lambda}(\theta_n^*)$ *and the strategy* $\theta_n^* = \arg\max_{\theta_n \in \Theta} E[\varphi(T_{n,\lambda}(\theta_n))]$ *when* $\varphi$ *is decreasing on* $(0, \infty)$ *as follows: for* $i = 1$*,* $\mathbb{P}(\vartheta_1^* = 1) = \mathbb{P}(\vartheta_1^* = 0) = 0.5$*, and for* $i \geq 2$*,*

$$\vartheta_i^* = \begin{cases} 1, & T_{i-1,\lambda}(\theta_{i-1}^*) \geq 0, \\ 0, & T_{i-1,\lambda}(\theta_{i-1}^*) < 0. \end{cases} \quad (10)$$

*Under the assumptions of RCM, we obtain*

$$E\left[|\varphi\left(T_{n,\lambda}(\theta_n^*)\right) - \varphi\left(\eta_n\right)|\right] = O\left(\frac{\sigma}{(1-\lambda)\sqrt{n}}\right), \quad (11)$$

*where* $\eta_n \sim \mathcal{B}\left(\omega_n, \sigma_0\right)$ *is a spike distribution with the parameter* $\omega_n = \lambda\mu/(1-\lambda) + \sqrt{n}\mu/\sigma$*,* $\sigma_0 = \sqrt{1 + \mu^2/\sigma^2}$*. The bandit distribution* $\mathcal{B}\left(\omega_n, \sigma_0\right)$ *has the density function*

$$f^{\omega_n, \sigma_0}(y) = \frac{1}{\sqrt{2\pi}\sigma_0} \exp\left(-\frac{(|y| - \omega_n)^2}{2\sigma_0^2}\right) \\ - \frac{\omega_n}{\sigma_0^2} e^{2\omega_n|y|/\sigma_0^2} \Phi\left(-\frac{|y| + \omega_n}{\sigma_0}\right). \quad (12)$$

It can be seen that in the specified probability density function $f^{\omega_n, \sigma_0}(y)$, the parameter $\omega_n$ exhibits a positive correlation with $\lambda$, $n$, and $\mu$, regulating the extent of the left-right shift of the dual peaks in the distribution. Figure 3 provides an example illustrating the effect of varying $\lambda$. Meanwhile, the parameter $\sigma_0$ influences the overall height of these peaks. When $\mu = 0$, the probability density function $f^{\omega_n, \sigma_0}(y)$ degenerates into the probability density function of the standard normal distribution.

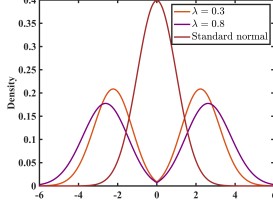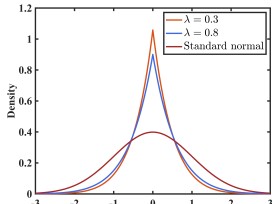

Figure 3: Density plots of asymptotic distributions under two hypotheses, compared with a standard normal distribution. The left graph is the density function under the alternative hypothesis. Intuitively, a larger $\lambda$ corresponding to a greater statistical power. The right graph is the density function when the null hypothesis holds. As illustrated, type I error can be effectively controlled.

**Theorem 4.2.** *Under aforementioned policy* $\theta_n^*$*, the oracle*

*test statistic $T_{n,\lambda}(\theta_n^*)$ satisfies*

$$\lim_{n\to\infty} \mathbb{P}\left(|T_{n,\lambda}(\theta_n^*)| > z_{1-\alpha/2}\right) = \Phi\left(\frac{\omega_n - z_{1-\alpha/2}}{\sigma_0}\right)$$
$$+ e^{\frac{2\omega_n z_{1-\alpha/2}}{\sigma_0^2}} \Phi\left(-\frac{\omega_n + z_{1-\alpha/2}}{\sigma_0}\right).$$

*Consequently, it attains the following properties:*

(i). (**Type I error control**): *Under the null hypothesis,*

$$\lim_{n\to\infty} \mathbb{P}\left(|T_{n,\lambda}(\theta_n^*)| > z_{1-\alpha/2}\right) \leq \alpha.$$

(ii). (**Consistency against fixed alternatives**): *For a given fixed $\mu > 0$,*

$$\lim_{n\to\infty} \mathbb{P}\left(|T_{n,\lambda}(\theta_n^*)| > z_{1-\alpha/2}\right)$$
$$= \lim_{n\to\infty} \sup_{\theta_n \in \Theta} \mathbb{P}\left(|T_{n,\lambda}(\theta_n)| > z_{1-\alpha/2}\right) = 1.$$

Theorem 4.2 demonstrates that $\theta_n^*$ is the optimal policy for hypothesis testing, as it ensures control over the type I error rate under the null hypothesis and achieves the maximum rejection probability under the alternative hypothesis. This finding indicates that $T_{n,\lambda}(\theta_n^*)$ reaches the highest level of statistical efficacy. To ensure validity under the non-oracle test, we impose the following assumptions.

**Assumption 1.** (Boundedness). (i) *There exists a scalar $\varepsilon > 0$ such that $\widehat{e}(X) \geq \varepsilon$. (ii) There exists some constant $M$ such that $\max(m_0(X), m_1(X)) \leq M$.*

**Assumption 2.** (Doubly Robust Specification). *At least one of the outcome regression functions and the propensity score function is correctly specified.*

The boundedness condition in Assumption 1 (i) is also referred to as the positivity assumption in the RCM. The double robustness condition in Assumption 2 can be replaced by requiring the outcome regression functions and the propensity score function to satisfy certain convergence rate conditions (Chernozhukov et al., 2018). Typically, machine learning algorithms satisfy the above convergence rate.

*Remark* 4.3. Corollary 2 of Liu & Xie (2020) demonstrates that the use of the Cauchy combination effectively mitigates the type I error rate. Moreover, Section 3 of Liu & Xie (2020) suggests that the statistical power is improved when the alternative hypothesis is true.

*Remark* 4.4. Under Assumption 2, we have $\mathbb{E}(\widehat{\mu}) = \mathbb{E}(Y^{(1)} - Y^{(0)})$, indicating that $T_{n,\lambda}(\theta_n^*)$ satisfies the properties of Theorem 4.1 and Theorem 4.2.

*Remark* 4.5. The choice of $K$ in cross fitting does not affect the asymptotic distribution of the estimator (Chernozhukov et al., 2018; Guo et al., 2021). The simulations in Section 5 show that a good performance is achievable with $K = 2$.

# 5. Experiments

In this section, we conduct detailed comparisons between the proposed method and other state-of-the-art methods via synthetic data (Section 5.1) and real-world data (Section 5.2).

### 5.1. Simulations studies

First, we conduct simulations studies to investigate the finite sample performance of the proposed method. Consider the following synthetic data-generating process:

The covariates $X_1$ and $X_2$ are two independent mean zero Gaussian distributions with $\text{var}(X_1) = \text{var}(X_2) = 1$. Consider a randomized controlled trial, where the value of $A$ is independent of any covariates and follows a Bernoulli distribution with a success probability $\mathbb{P}(A = 1) = 0.5$. The outcome $Y$ is generated by $Y = F(X_1, X_2) + A \cdot G(X_1, X_2) + \varepsilon$, where $\varepsilon$ is a Gaussian noise with a mean of zero and a standard deviation $\sigma_\varepsilon$.

Table 1: Functions and standard deviation considered in the simulation studies.

| | $F(X_1, X_2)$ | $G(X_1, X_2)$ | $\sigma_\varepsilon$ |
|---|---|---|---|
| I | $2X_1 + X_2$ | $0$ | 0.5 |
| II | $X_1(X_2 + 1)$ | $(X_1 + 2X_2)/10$ | |
| III | $X_1^2 + X_2 + 1$ | $(X_1^2 + X_2^2)/110$ | 0.6 |
| IV | $0.5X_1 e^{X_2}$ | $(X_1 + 2X_2^2)/105$ | |

Table 1 presents a more detailed illustration of the configurations. We consider four different functions $F(X_1, X_2)$ and $G(X_1, X_2)$, including both linear and nonlinear forms with two different values for standard deviations $\sigma_\varepsilon$. These configurations collectively result in a total of 32 distinct simulation settings. The initial two functions $G_{\text{I}}(X_1, X_2)$ and $G_{\text{II}}(X_1, X_2)$ represent scenarios where the null hypothesis is true, while the latter two functions $G_{\text{III}}(X_1, X_2)$ and $G_{\text{IV}}(X_1, X_2)$ correspond to cases where the alternative hypothesis holds. Note that $G_{\text{II}}$, $G_{\text{III}}$, and $G_{\text{IV}}$ are all heterogeneous, meaning that the treatment effect varies with the covariates. The effect of treatment is positive for all samples in $G_{\text{III}}$, differing only in magnitude. However, a portion of the samples exhibit negative treatment effects in $G_{\text{IV}}$, complicating the accurate testing of ATE. The empirical type I error rates and power of five distinct methods are evaluated: (i) Permuted WTAB, denoted by PWTAB; (ii) Weighted Two-Arm Bandit process, denoted by WTAB; (iii) $z$-test based on DML, denoted by $z$-DML; (iv) CUPED; (v) Difference-in-mean estimator, denoted by DIM, and the sample size is fixed to $n = 20000$.

The results for both the null and alternative hypotheses are presented in Table 2 and Figure 4, respectively. As evidenced by the results presented in Table 2, the five methods

Table 2: Type I error rates of five different statistics when null hypothesis holds.

| $\mathcal{H}_0$ | | $G_{\mathrm{I}}(X_1, X_2)$ | | | | | $G_{\mathrm{II}}(X_1, X_2)$ | | | | |
|---|---|---|---|---|---|---|---|---|---|---|---|
| $\sigma_\varepsilon$ | $F(X_1, X_2)$ | PWTAB | WTAB | $z$-DML | CUPED | DIM | PWTAB | WTAB | $z$-DML | CUPED | DIM |
| | I | 0.046 | 0.052 | 0.042 | 0.034 | 0.052 | 0.030 | 0.030 | 0.022 | 0.028 | 0.032 |
| 0.5 | II | 0.046 | 0.050 | 0.042 | 0.040 | 0.040 | 0.056 | 0.054 | 0.052 | 0.053 | 0.056 |
| | III | 0.054 | 0.056 | 0.042 | 0.042 | 0.036 | 0.046 | 0.052 | 0.042 | 0.032 | 0.044 |
| | IV | 0.044 | 0.046 | 0.040 | 0.054 | 0.038 | 0.046 | 0.042 | 0.040 | 0.056 | 0.042 |
| | I | 0.044 | 0.042 | 0.042 | 0.044 | 0.060 | 0.060 | 0.056 | 0.060 | 0.050 | 0.058 |
| 0.6 | II | 0.040 | 0.048 | 0.036 | 0.028 | 0.042 | 0.040 | 0.050 | 0.032 | 0.054 | 0.038 |
| | III | 0.044 | 0.052 | 0.044 | 0.070 | 0.060 | 0.048 | 0.050 | 0.046 | 0.030 | 0.054 |
| | IV | 0.052 | 0.052 | 0.044 | 0.056 | 0.048 | 0.050 | 0.050 | 0.042 | 0.024 | 0.032 |

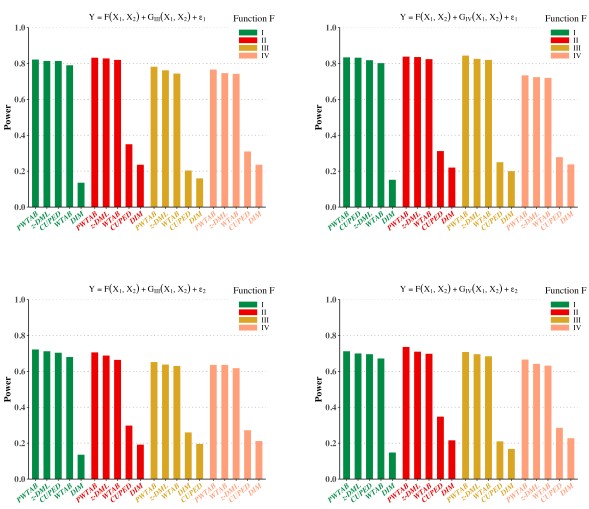

Figure 4: Power comparisons of various methods across different settings, presented in descending order, with PWTAB consistently demonstrating the best performance.

demonstrate the capacity to control the type I error rate under a range of conditions, including different functions and noise. In terms of power, Figure 4 presents a comparison of different methods, with fixed $G(X_1, X_2)$ and $\sigma_\varepsilon$ in each subfigure. When confronted with a linear function $F(X_1, X_2)$, CUPED, $z$-DML, WTAB, and PWTAB exhibit comparable power, and PWTAB shows superior performance. In the case of a nonlinear function, the efficacy of CUPED is significantly reduced. In contrast, $z$-DML, WTAB, and PWTAB retain their effectiveness, demonstrating greater adaptability to complex functions, greater robustness, and improved power. Furthermore, PWTAB consistently outperforms $z$-DML, since WTAB and $z$-DML exhibit comparable power in almost all cases. As a result, a higher statistical power is achieved after the aggregation of the $p$-values. It is important to emphasize that the similarity in performance between WTAB and the test $z$ does not contradict our theory. In fact,

we construct $T_{n,\lambda}(\theta_n)$ with the objective of maximizing tail probabilities, ensuring that $T_{n,\lambda}(\theta_n^*)$ achieves the largest statistical power in one-sided hypothesis testing with a two-tailed rejection region.

**More ML-based simulation studies**. As discussed previously, additional simulation studies were performed to evaluate the effectiveness of the proposed method, including the use of another machine learning algorithm XGBoost and the stacking of the ensemble learning approach. A more detailed comparison can be found in the Appendix A, and PWTAB consistently outperforms other methods.

### 5.2. Real Data Analysis

In this section, the application of the proposed method is demonstrated through an analysis of three real data sets obtained from a world-leading ride-sharing company. Due to privacy considerations, we refer to them as data sets A, B, and C. The company leverages different exposures within the app to incentivize users to consume. Specifically, the dataset is collected through a randomized controlled trial. The company randomly decides whether or not to expose new marketing strategies to target consumers. To improve returns, it is crucial for the company to accurately assess the

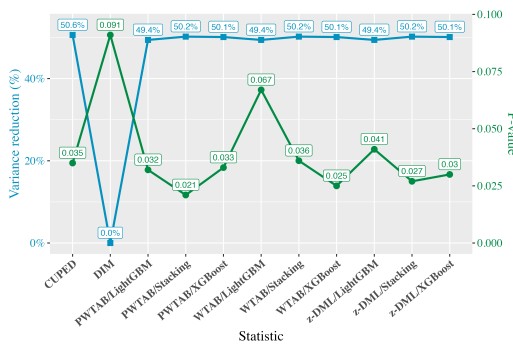

Figure 5: Variance reduction compared to DIM (left $y$-axis) and $p$-values of statistics (right $y$-axis).

long-term impacts of various strategies.

The numerical results are reported in Figure 5 and Table 3. It can be seen in Figure 5 that, compared to DIM, both the proposed method and CUPED exhibit a comparable reduction in variance. However, PWTAB yields smaller $p$-values regardless of the machine learning algorithm used, indicating that PWTAB offers superior statistical power. Additionally, the use of stacking allows for the integration of the strengths of different machine learning algorithms, leading to more accurate and robust results. This conclusion is further supported by the results presented in Table 3 (a), where the $p$-value of CUPED fails to reach the threshold necessary to reject the null hypothesis, whereas PWTAB successfully meets this criterion, demonstrating its superior efficacy.

Table 3: Hypothesis testing $p$-values obtained based on different machine learning algorithms and statistics for real-world datasets.

(a) $p$-values on dataset B

| Method | PWTAB | WTAB | $z$-DML | CUPED | DIM |
|---|---|---|---|---|---|
| LightGBM | 0.044 | 0.043 | 0.055 | 0.053 | 0.228 |
| XGBoost | 0.044 | 0.069 | 0.052 | 0.053 | 0.228 |
| Stacking | 0.032 | 0.055 | 0.053 | 0.053 | 0.228 |

(b) $p$-values on dataset C

| Method | PWTAB | WTAB | $z$-DML | CUPED | DIM |
|---|---|---|---|---|---|
| LightGBM | 0.837 | 0.839 | 0.713 | 0.745 | 0.850 |
| XGBoost | 0.909 | 0.491 | 0.726 | 0.745 | 0.850 |
| Stacking | 0.807 | 0.827 | 0.731 | 0.745 | 0.850 |

Given that real-world data distributions are often difficult to replicate using purely synthetic data, we additionally construct a semi-synthetic dataset based on real-world data. Following the approach proposed in (Kohavi et al., 2020), we generate synthetic data based on real-world observations. The corresponding numerical results are presented in Table 4. We can see that the proposed PWTAB method effectively controls type I error while achieving the highest statistical power among all compared methods, even in scenarios involving high-variance data or challenging distributional characteristics. This demonstrates that the proposed statistic substantially enhances the sensitivity of the A/B testing and exhibits strong robustness.

## 6. Conclusion and Future Works

In this paper, we propose a new maximum probability-driven TAB process by weighting the mean volatility statistic for a more powerful A/B testing. The proposed method constructs counterfactual results through doubly robust esti-

Table 4: Type I error rates and statistical power based on synthetic data derived from real-world dataset.

| Method | Metric | PWTAB | WTAB | $z$-DML | CUPED | DIM |
|---|---|---|---|---|---|---|
| LightGBM | Type I Error | 0.052 | 0.052 | 0.044 | 0.050 | 0.048 |
| | Power | 0.758 | 0.738 | 0.744 | 0.740 | 0.498 |
| XGBoost | Type I Error | 0.052 | 0.034 | 0.046 | 0.050 | 0.048 |
| | Power | 0.758 | 0.738 | 0.746 | 0.740 | 0.498 |
| Stacking | Type I Error | 0.052 | 0.052 | 0.046 | 0.050 | 0.048 |
| | Power | 0.764 | 0.732 | 0.746 | 0.740 | 0.498 |

mation, controls the type I error rate, and improves statistical power by incorporating weights and permutation. Our numerical results in both simulation and real-world data analysis demonstrate the effectiveness of the proposed approach. By increasing the sensitivity of randomized controlled trials, our method allows for more precise value assessments and the ability to conduct experiments on smaller populations. Although machine learning provides a general approach to capture the distribution characteristics of data, it is not refined enough compared to explicit models which should be explored in the ride-sharing company. We leave the task of modeling the data distribution to future work.

## Acknowledgements

We thank the anonymous referees and the meta reviewer for their constructive comments, which have led to a significant improvement of the earlier version of this article. Xiaodong Yan was supported by the National Key R&D Program of China (No. 2023YFA1008701) and the National Natural Science Foundation of China (No. 12371292) and CCF-DiDi GAIA Collaborative Research Funds for Young Scholars.

## Impact Statement

This paper presents work whose goal is to improve the sensitivity, shorten the cycle time and reduce the cost of A/B testing. There are many potential societal consequences of our work, none which we feel must be specifically highlighted here.

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

# A. Additional experiments

Table 5 presents the empirical type I error rates for all methods in the synthetic data-based experiment. It further validates the effectiveness of the proposed methods, with findings aligning closely with those in Section 5. Specifically, under both sharp null hypothesis and null hypothesis with heterogeneous treatment effects, as well as different variance of noise, the proposed method demonstrates robust control of type I error rates, effectively preventing overestimation of the average treatment effect.

Table 5: More simulation results of type I error rates of five different statistics when null hypothesis holds.

| | | $\mathcal{H}_0$ | $G_{\mathrm{I}}(X_1, X_2)$ | | | | | $G_{\mathrm{II}}(X_1, X_2)$ | | | | |
|---|---|---|---|---|---|---|---|---|---|---|---|---|
| Methods | $\sigma_\varepsilon$ | $F(X_1, X_2)$ | PWTAB | WTAB | $z$-DML | CUPED | DIM | PWTAB | WTAB | $z$-DML | CUPED | DIM |
| | | I | 0.046 | 0.052 | 0.042 | 0.034 | 0.052 | 0.030 | 0.030 | 0.022 | 0.028 | 0.032 |
| LightGBM | 0.5 | II | 0.046 | 0.050 | 0.042 | 0.040 | 0.040 | 0.056 | 0.054 | 0.052 | 0.053 | 0.056 |
| | | III | 0.054 | 0.056 | 0.042 | 0.042 | 0.036 | 0.046 | 0.052 | 0.042 | 0.032 | 0.044 |
| | | IV | 0.044 | 0.046 | 0.040 | 0.054 | 0.038 | 0.046 | 0.042 | 0.040 | 0.056 | 0.042 |
| | | I | 0.044 | 0.042 | 0.042 | 0.044 | 0.060 | 0.060 | 0.056 | 0.060 | 0.050 | 0.058 |
| LightGBM | 0.6 | II | 0.040 | 0.048 | 0.036 | 0.028 | 0.042 | 0.040 | 0.050 | 0.032 | 0.054 | 0.038 |
| | | III | 0.044 | 0.052 | 0.044 | 0.070 | 0.060 | 0.048 | 0.050 | 0.046 | 0.030 | 0.054 |
| | | IV | 0.052 | 0.052 | 0.044 | 0.056 | 0.048 | 0.050 | 0.050 | 0.042 | 0.024 | 0.032 |
| | | I | 0.052 | 0.036 | 0.046 | 0.034 | 0.052 | 0.032 | 0.042 | 0.026 | 0.028 | 0.032 |
| XGBoost | 0.5 | II | 0.050 | 0.050 | 0.044 | 0.040 | 0.040 | 0.044 | 0.050 | 0.042 | 0.052 | 0.056 |
| | | III | 0.046 | 0.052 | 0.044 | 0.042 | 0.036 | 0.052 | 0.046 | 0.048 | 0.032 | 0.044 |
| | | IV | 0.038 | 0.044 | 0.032 | 0.054 | 0.038 | 0.058 | 0.060 | 0.056 | 0.056 | 0.042 |
| | | I | 0.048 | 0.046 | 0.040 | 0.044 | 0.060 | 0.054 | 0.062 | 0.048 | 0.050 | 0.058 |
| XGBoost | 0.6 | II | 0.042 | 0.044 | 0.040 | 0.028 | 0.042 | 0.044 | 0.026 | 0.036 | 0.054 | 0.038 |
| | | III | 0.068 | 0.058 | 0.058 | 0.070 | 0.060 | 0.046 | 0.054 | 0.042 | 0.030 | 0.054 |
| | | IV | 0.064 | 0.062 | 0.046 | 0.056 | 0.048 | 0.044 | 0.044 | 0.040 | 0.024 | 0.032 |
| | | I | 0.044 | 0.040 | 0.042 | 0.034 | 0.052 | 0.032 | 0.032 | 0.028 | 0.028 | 0.032 |
| Stacking | 0.5 | II | 0.056 | 0.048 | 0.048 | 0.040 | 0.040 | 0.054 | 0.048 | 0.044 | 0.052 | 0.056 |
| | | III | 0.054 | 0.046 | 0.046 | 0.042 | 0.036 | 0.056 | 0.050 | 0.040 | 0.032 | 0.044 |
| | | IV | 0.044 | 0.040 | 0.038 | 0.054 | 0.038 | 0.058 | 0.060 | 0.050 | 0.056 | 0.042 |
| | | I | 0.044 | 0.048 | 0.040 | 0.044 | 0.060 | 0.060 | 0.056 | 0.060 | 0.050 | 0.058 |
| Stacking | 0.6 | II | 0.040 | 0.044 | 0.038 | 0.028 | 0.042 | 0.026 | 0.038 | 0.028 | 0.054 | 0.038 |
| | | III | 0.054 | 0.052 | 0.048 | 0.070 | 0.060 | 0.054 | 0.062 | 0.044 | 0.030 | 0.054 |
| | | IV | 0.060 | 0.066 | 0.048 | 0.056 | 0.048 | 0.046 | 0.046 | 0.040 | 0.024 | 0.032 |

Figure 6 presents the power comparison for all methods in the synthetic data-based. The comparative analysis reveals two key findings regarding the performance characteristics. Horizontally, under the same machine learning algorithms, PWTAB consistently demonstrates superior statistical power, indicating its optimal performance. Vertically, when employing fixed statistics, the ensemble learning-based stacking approach effectively integrates the strengths of different machine learning algorithms, achieving comparable or even enhanced performance relative to the current best methods.

# B. Asymptotic distribution and Proofs

Let $\{B_s\}_{s \geq 0}$ be the standard Brownian motion on $(\Omega, \mathcal{F}, P)$ and $(\mathcal{F}_s^*)_{s \geq 0}$ be the natural filtration generated by $\{B_s\}_{s \geq 0}$.

For any integer $m \geq 1$, let $C_b^m(\mathbb{R})$ denote the set of functions on $\mathbb{R}$ that have bounded derivatives up to order $m$. Let $\varphi \in C_b^3(\mathbb{R})$ be an even function, for any $\delta \in \mathbb{R}$, $\beta > 0$ and $t \in [0, 1)$, we define $F_1(x) = \varphi(x)$, and

$$F_t(x) = \int_{\mathbb{R}} \varphi(z) q_{\delta,\beta}(t, x, z) \mathrm{d}z, \tag{13}$$

where

$$q_{\delta,\beta}(t, x, z) = \frac{1}{\beta\sqrt{2\pi(1-t)}} e^{-\frac{(x-z)^2 - 2\delta(1-t)(|z|-|x|) + \delta^2(1-t)^2}{2(1-t)\beta^2}} - \frac{\delta}{\beta^2} e^{\frac{2\delta|z|}{\beta^2}} \Phi\left(-\frac{|z| + |x| + \delta(1-t)}{\beta\sqrt{1-t}}\right).$$

Here, the dependence of $F_t$ on $\varphi$, $\delta$, and $\beta$ is not explicitly noted for simplicity. It is clear from the definition that

$$F_0(0) = E[\varphi(\eta)],$$

where $\eta \sim \mathcal{B}(\delta, \beta)$ is a spike distribution.

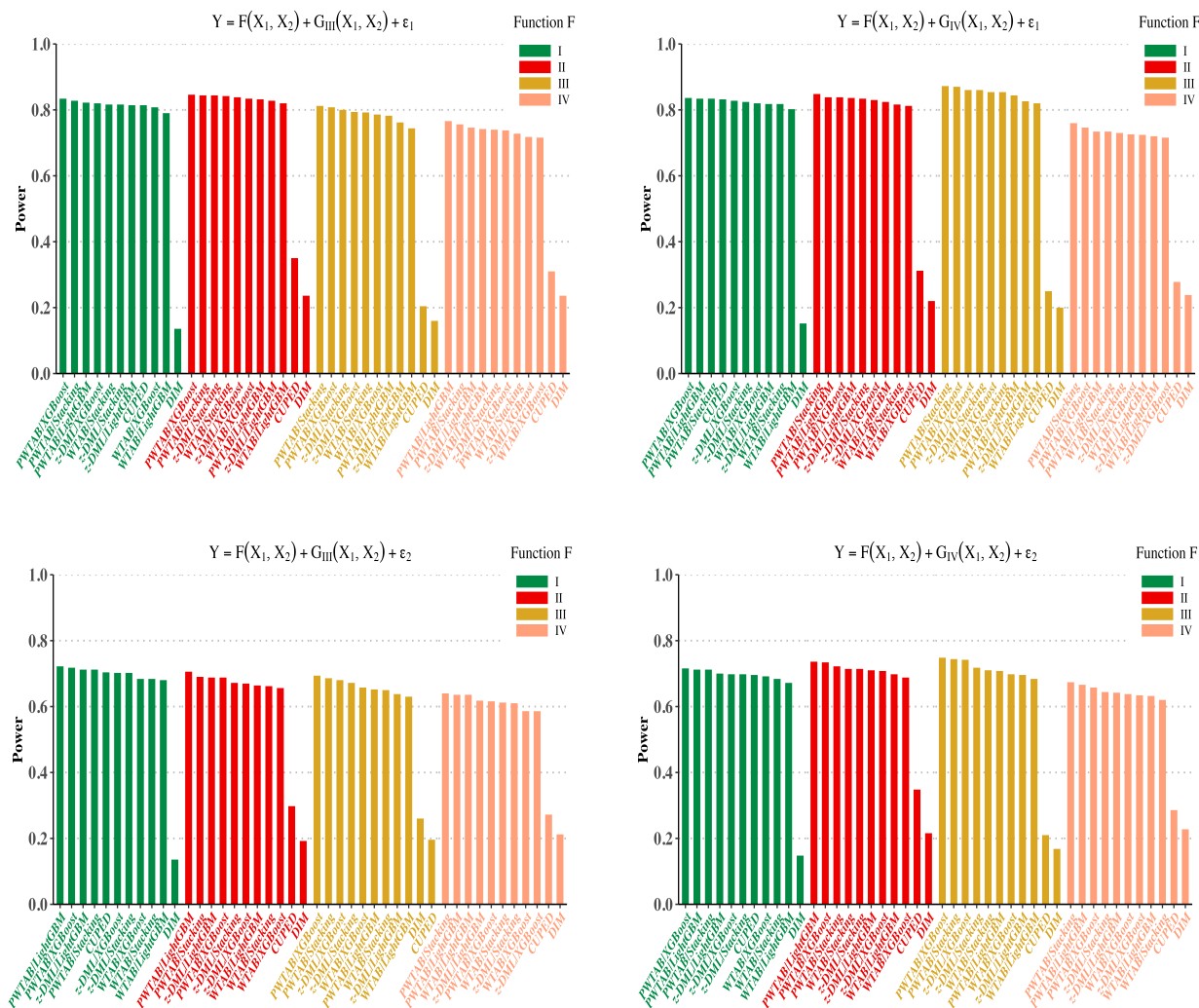

Figure 6: More power comparisons of various methods across different settings, presented in descending order.

The following lemma lists some analytic properties of the family $\{F_t(x)\}_{t \in [0,1]}$.

**Lemma B.1.** *Let the number of dots on top of a function denote the same order derivatives with respect to $x$.*

*(1) For each fixed $t \in [0,1]$, $F_t(x) \in C_b^2(\mathbb{R})$. In addition, the first and second order derivatives of $F_t(x)$ are uniformly bounded for all $0 \le t \le 1$ and $x$.*

*(2) The family $\{\ddot{F}_t(x)\}_{t \in [0,1]}$ is uniformly Lipschitz, i.e., there exists a constant $L$, independent with $t$, such that*

$$\left| \ddot{F}_t(x_1) - \ddot{F}_t(x_2) \right| \le L \left| x_1 - x_2 \right|, \quad x_1, x_2 \in \mathbb{R}.$$

*(3) For any $t \in [0,1]$, $F_t(x)$ is an even function. Furthermore, if for any $x \in \mathbb{R}$,*

$$\operatorname{sgn}(\dot{\varphi}(x)) = \pm \operatorname{sgn}(x),$$

*then*

$$\text{sgn}(\dot{F}_t(x)) = \pm \text{sgn}(x), x \in \mathbb{R}.$$

*(4) If* $\text{sgn}(\dot{\varphi}(x)) = \pm \text{sgn}(x)$ *for all* $x \in \mathbb{R}$, *then*

$$\sum_{m=1}^{n} \sup_{x \in \mathbb{R}} \left| F_{\frac{m-1}{n}}(x) - F_{\frac{m}{n}}(x) \mp \frac{\delta}{n} \left| \dot{F}_{\frac{m}{n}}(x) \right| - \frac{\beta^2}{2n} \ddot{F}_{\frac{m}{n}}(x) \right| = O\left( \frac{\beta|\delta|}{n} + \frac{\beta}{\sqrt{n}} \right).$$

*Proof.* We prove the lemma in numerical order.

(1) For $t = 1$, $F_1(x) \equiv \varphi(x)$ and the result is trivial.

Next, we assume that $0 \leq t < 1$. Since $\varphi$ is an even function, with the definition of $F_t(x)$, it follows by direct calculation that

$$\dot{F}_t(x) = \int_0^{\infty} \frac{sgn(x)}{\beta\sqrt{2\pi(1-t)}} \dot{\varphi}(z) \mathrm{e}^{-\frac{(z-\delta(1-t)-|x|)^2}{2(1-t)\beta^2}} \left[ 1 - \mathrm{e}^{-\frac{2|x|z}{(1-t)\beta^2}} \right] \mathrm{d}z,$$

$$\ddot{F}_t(x) = \int_0^{\infty} \frac{1}{\beta\sqrt{2\pi(1-t)}} \ddot{\varphi}(z) \mathrm{e}^{-\frac{(z-\delta(1-t)-|x|)^2}{2(1-t)\beta^2}} \left[ 1 + \mathrm{e}^{-\frac{2|x|z}{(1-t)\beta^2}} \right] \mathrm{d}z$$

$$+ \int_0^{\infty} \frac{2\delta}{\beta^2\sqrt{2\pi(1-t)}} \dot{\varphi}(z) \mathrm{e}^{-\frac{(z+\delta(1-t)+|x|)^2}{2(1-t)\beta^2}} \mathrm{e}^{\frac{2\delta z}{\beta^2}} \mathrm{d}z$$

$$= \int_0^{\infty} \frac{1}{\beta\sqrt{2\pi(1-t)}} \ddot{\varphi}(z) \mathrm{e}^{-\frac{(z-\delta(1-t)-|x|)^2}{2(1-t)\beta^2}} \left[ 1 + \mathrm{e}^{-\frac{2|x|z}{(1-t)\beta^2}} \right] \mathrm{d}z$$

$$+ \int_0^{\infty} \frac{2\delta}{\beta^2\sqrt{2\pi(1-t)}} \dot{\varphi}(z) \mathrm{e}^{-\frac{(z-\delta(1-t)+|x|)^2}{2(1-t)\beta^2}} \mathrm{e}^{-\frac{2\delta|x|}{\beta^2}} \mathrm{d}z. \tag{14}$$

Since $\varphi \in C_b^3(\mathbb{R})$, we conclude that $F_t(x) \in C_b^2(\mathbb{R})$, and the first- and second-order derivatives of $F_t(x)$ are uniformly bounded for all $t$ and $x$.

(2) For $x < 0$, we have

$$\dddot{F}(x) = \int_0^{\infty} \frac{1}{\beta\sqrt{2\pi(1-t)}} \dddot{\varphi}(z) \mathrm{e}^{-\frac{(z-\delta(1-t)+x)^2}{2(1-t)\beta^2}} \left[ \mathrm{e}^{\frac{2xz}{(1-t)\beta^2}} - 1 \right] \mathrm{d}z$$

$$+ \int_0^{\infty} \frac{4\delta}{\beta^3\sqrt{2\pi(1-t)}} [\delta\dot{\varphi}(z) + \beta\ddot{\varphi}(z)] \mathrm{e}^{-\frac{(z+\delta(1-t)-x)^2}{2(1-t)\beta^2}} \mathrm{e}^{\frac{2\delta z}{\beta^2}} \mathrm{d}z$$

$$= \int_0^{\infty} \frac{1}{\beta\sqrt{2\pi(1-t)}} \dddot{\varphi}(z) \mathrm{e}^{-\frac{(z-\delta(1-t)+x)^2}{2(1-t)\beta^2}} \left[ \mathrm{e}^{\frac{2xz}{(1-t)\beta^2}} - 1 \right] \mathrm{d}z$$

$$+ \int_0^{\infty} \frac{4\delta}{\beta^3\sqrt{2\pi(1-t)}} [\delta\dot{\varphi}(z) + \beta\ddot{\varphi}(z)] \mathrm{e}^{-\frac{(z-\delta(1-t)-x)^2}{2(1-t)\beta^2}} \mathrm{e}^{\frac{2\delta x}{\beta^2}} \mathrm{d}z.$$

For $x > 0$, we have

$$\dddot{F}_t(x) = \int_0^{\infty} \frac{1}{\beta\sqrt{2\pi(1-t)}} \dddot{\varphi}(z) \mathrm{e}^{-\frac{(z-\delta(1-t)-x)^2}{2(1-t)\beta^2}} \left[ 1 - \mathrm{e}^{-\frac{2xz}{(1-t)\beta^2}} \right] \mathrm{d}z$$

$$- \int_0^{\infty} \frac{4\delta}{\beta^3\sqrt{2\pi(1-t)}} [\beta\ddot{\varphi}(z) + \delta\dot{\varphi}(z)] \mathrm{e}^{-\frac{(z+\delta(1-t)+x)^2}{2(1-t)\beta^2}} \mathrm{e}^{\frac{2\delta z}{\beta^2}} \mathrm{d}z$$

$$= \int_0^{\infty} \frac{1}{\beta\sqrt{2\pi(1-t)}} \dddot{\varphi}(z) \mathrm{e}^{-\frac{(z-\delta(1-t)-x)^2}{2(1-t)\beta^2}} \left[ 1 - \mathrm{e}^{-\frac{2xz}{(1-t)\beta^2}} \right] \mathrm{d}z$$

$$- \int_0^{\infty} \frac{4\delta}{\beta^3\sqrt{2\pi(1-t)}} [\beta\ddot{\varphi}(z) + \delta\dot{\varphi}(z)] \mathrm{e}^{-\frac{(z-\delta(1-t)+x)^2}{2(1-t)\beta^2}} \mathrm{e}^{-\frac{2\delta x}{\beta^2}} \mathrm{d}z.$$

Since $\varphi \in C_b^3(\mathbb{R})$, it follows that $\dddot{F}_t(x)$ is uniformly bounded for all $t$ and $x \neq 0$. For $x = 0$, the third-order left and right derivatives of $F_t(x)$ can be shown to exist and are also bounded uniformly in $t$. Thus by the mean value theorem, one can

find a constant $L$, independent with $t$, such that for any $x_1, x_2 \in \mathbb{R}$,

$$\left| \ddot{F}_t(x_1) - \ddot{F}_t(x_2) \right| \le L |x_1 - x_2|.$$

(3) It follows by direct calculation that for any $x \in \mathbb{R}$,

$$
\begin{aligned}
F_t(x) &= \int_{\mathbb{R}} \varphi(z) q_{\delta,\beta}(t,x,z)\mathrm{d}z = \int_{\mathbb{R}} \varphi(z) q_{\delta,\beta}(t,-x,-z)\mathrm{d}z \\
&= \int_{\mathbb{R}} \varphi(z) q_{\delta,\beta}(t,-x,z)\mathrm{d}z \\
&= F_t(-x).
\end{aligned}
$$

That is $F_t$ is an even function. By (14) we have that for any $x \in \mathbb{R}$,

$$\operatorname{sgn}\left( \dot{F}_t(x) \right) = \pm \operatorname{sgn}(x) \quad \text{when } \operatorname{sgn}\left( \dot{\varphi}(x) \right) = \pm \operatorname{sgn}(x).$$

(4) We only prove the case $\operatorname{sgn}\left( \dot{\varphi}(x) \right) = \operatorname{sgn}(x)$. The other case follows by similar arguments.

For any $(t,x) \in [0,1] \times \mathbb{R}$, let $\{Y_s^{t,x}\}_{s \in [t,1]}$ denote the solution of the SDE

$$
\begin{cases}
\mathrm{d}Y_s^{t,x} = \frac{\delta}{\beta} \operatorname{sgn}\left( Y_s^{t,x} \right) \mathrm{d}s + \mathrm{d}B_s, & s \in [t,1] \\
Y_t^{t,x} = x.
\end{cases}
\tag{15}
$$

Although the drift coefficient is discontinuous, this equation does have a unique strong solution (see Mel'nikov (1979), Theorem 1). Fortunately, $\{Y_s^{t,x}\}_{s \in [t,1]}$ has an explicit probability density function, which can be denoted by

$$q_{\frac{\delta}{\beta}}(t,x;s,z) = \frac{1}{\sqrt{2\pi(s-t)}} \mathrm{e}^{-\frac{(x-z)^2 - 2\delta(s-t)(|z|-|x|)/\beta + \delta^2(s-t)^2/\beta^2}{2(s-t)}} - \frac{\delta}{\beta} \mathrm{e}^{\frac{2\delta|z|}{\beta}} \int_{|x|+|z|+\delta(s-t)/\beta}^{\infty} \frac{1}{\sqrt{2\pi(s-t)}} \mathrm{e}^{-\frac{u^2}{2(s-t)}} \mathrm{d}u.$$

Then, the basic function $F_t$ can also be denoted by

$$F_t(x) = \mathbb{E}\left[ \varphi\left( \beta Y_1^{t,\frac{x}{\beta}} \right) \right].
\tag{16}$$

Follows from the Markov property of $(Y_s^{t,x})$, we have for any $b \in [0, 1-t]$,

$$F_t(x) = \mathbb{E}\left[ \varphi\left( \beta Y_1^{t,\frac{x}{\beta}} \right) \right] = \mathbb{E}\left[ \mathbb{E}\left[ \varphi\left( \beta Y_1^{t,\frac{x}{\beta}} \right) | \mathcal{F}_{t+h}^* \right] \right] = \mathbb{E}\left[ F_{t+h}\left( \beta Y_{t+b}^{t,\frac{x}{\beta}} \right) \right].$$

Applying the Markov property and (16), we have for any $1 \le m \le n$,

$$F_{\frac{m-1}{n}}(x) = \mathbb{E}\left[ F_{\frac{m}{n}}\left( \beta Y_{\frac{m}{n}}^{\frac{m-1}{n},\frac{x}{\beta}} \right) \right].$$

By Itô's formula, we have

$$F_{\frac{m}{n}}\left( \beta Y_{\frac{m}{n}}^{\frac{m-1}{n},\frac{x}{\beta}} \right) = F_{\frac{m}{n}}(x) + \int_{\frac{m-1}{n}}^{\frac{m}{n}} \dot{F}_{\frac{m}{n}}\left( \beta Y_s^{\frac{m-1}{n},\frac{x}{\beta}} \right) \beta \mathrm{d}Y_s^{\frac{m-1}{n},\frac{x}{\beta}} + \frac{\beta^2}{2} \int_{\frac{m-1}{n}}^{\frac{m}{n}} \ddot{F}_{\frac{m}{n}}\left( \beta Y_s^{\frac{m-1}{n},\frac{x}{\beta}} \right) \mathrm{d}s.$$

This combined with (3) implies that

$$
\begin{aligned}
F_{\frac{m-1}{n}}(x) &= \mathbb{E}\left[ F_{\frac{m}{n}}(x) + \int_{\frac{m-1}{n}}^{\frac{m}{n}} \dot{F}_{\frac{m}{n}}\left( \beta Y_s^{\frac{m-1}{n},\frac{x}{\beta}} \right) \beta \mathrm{d}Y_s^{\frac{m-1}{n},\frac{x}{\beta}} + \frac{\beta^2}{2} \int_{\frac{m-1}{n}}^{\frac{m}{n}} \ddot{F}_{\frac{m}{n}}\left( \beta Y_s^{\frac{m-1}{n},\frac{x}{\beta}} \right) \mathrm{d}s \right] \\
&= \mathbb{E}\left[ F_{\frac{m}{n}}(x) + \int_{\frac{m-1}{n}}^{\frac{m}{n}} \delta \dot{F}_{\frac{m}{n}}\left( \beta Y_s^{\frac{m-1}{n},\frac{x}{\beta}} \right) \operatorname{sgn}\left( \beta Y_s^{\frac{m-1}{n},\frac{x}{\beta}} \right) \mathrm{d}s + \frac{\beta^2}{2} \int_{\frac{m-1}{n}}^{\frac{m}{n}} \ddot{F}_{\frac{m}{n}}\left( \beta Y_s^{\frac{m-1}{n},\frac{x}{\beta}} \right) \mathrm{d}s \right] \\
&= \mathbb{E}\left[ F_{\frac{m}{n}}(x) + \int_{\frac{m-1}{n}}^{\frac{m}{n}} \delta \left| \dot{F}_{\frac{m}{n}}\left( \beta Y_s^{\frac{m-1}{n},\frac{x}{\beta}} \right) \right| \mathrm{d}s + \frac{\beta^2}{2} \int_{\frac{m-1}{n}}^{\frac{m}{n}} \ddot{F}_{\frac{m}{n}}\left( \beta Y_s^{\frac{m-1}{n},\frac{x}{\beta}} \right) \mathrm{d}s \right].
\end{aligned}
$$

Taking the supremum over $x$, we obtain

$$\sum_{m=1}^{n} \sup_{x \in \mathbb{R}} \left| F_{\frac{m-1}{n}}(x) - F_{\frac{m}{n}}(x) - \frac{\delta}{n} \left| \dot{F}_{\frac{m}{n}}(x) \right| - \frac{\beta^2}{2n} \ddot{F}_{\frac{m}{n}}(x) \right|$$

$$\leq \sum_{m=1}^{n} \sup_{x \in \mathbb{R}} \mathbb{E} \left[ \int_{\frac{m-1}{n}}^{\frac{m}{n}} |\delta| \left| \dot{F}_{\frac{m}{n}} \left( \beta Y_s^{\frac{m-1}{n}, \frac{x}{\beta}} \right) - \dot{F}_{\frac{m}{n}}(x) \right| \mathrm{d}s + \frac{1}{2} \int_{\frac{m-1}{n}}^{\frac{m}{n}} \left| \ddot{F}_{\frac{m}{n}} \left( \beta Y_s^{\frac{m-1}{n}, \frac{x}{\beta}} \right) - \ddot{F}_{\frac{m}{n}}(x) \right| \mathrm{d}s \right]$$

$$\leq \sum_{m=1}^{n} \sup_{x \in \mathbb{R}} \frac{C}{n} \mathbb{E} \left[ \sup_{s \in \left[ \frac{m-1}{n}, \frac{m}{n} \right]} \left| \beta Y_s^{\frac{m-1}{n}, \frac{x}{\beta}} - x \right| \right]$$

$$\leq \sum_{m=1}^{n} \frac{C\beta}{n} \mathbb{E} \left[ \frac{|\delta|}{n} + \sup_{s \in \left[ \frac{m-1}{n}, \frac{m}{n} \right]} \left| B_s - B_{\frac{m-1}{n}} \right| \right]$$

$$\leq C\beta \left( \frac{|\delta|}{n} + \frac{1}{\sqrt{n}} \right),$$

where $C$ is a constant depending only on $\delta$, $L$ and the bound of $\ddot{F}_t(x)$. This concludes the proof of the lemma. $\qquad \square$

All results below are under the assumptions Theorems 4.1 and 4.2.

**Lemma B.2.** *Let $\varphi \in C_b^3(\mathbb{R})$ be symmetric with centre $c \in \mathbb{R}$, and $\{F_t(x)\}_{t \in [0,1]}$ be defined as in (13). For any $\theta_n \in \Theta$, $n \in \mathbb{N}^+$ and $1 \leq m \leq n$, set*

$$\Gamma(m, n, \theta_n) = F_{\frac{m}{n}}(T_{m-1,\lambda}(\theta_n)) + \dot{F}_{\frac{m}{n}}(T_{m-1,\lambda}(\theta_n)) \left( \frac{\lambda \bar{R}_m^{(\vartheta_m)}}{(1-\lambda)n} + \frac{R_m^{(\vartheta_m)}}{\sqrt{n}\hat{\sigma}} \right) + \frac{1}{2} \ddot{F}_{\frac{m}{n}}(T_{m-1,\lambda}(\theta_n)) \left( \frac{R_m^{(\vartheta_m)}}{\sqrt{n}\hat{\sigma}} \right)^2. \quad (17)$$

*Then, we have*

$$\sum_{m=1}^{n} \mathbb{E} \left[ \left| F_{\frac{m}{n}}(T_{m,\lambda}(\theta_n)) - \Gamma(m, n, \theta_n) \right| \right] = O \left( \frac{\sigma}{(1-\lambda)\sqrt{n}} \right). \quad (18)$$

*Proof.* In fact, by (1) and (2) of Lemma B.1, there exists a constant $C > 0$ such that

$$\sup_{t \in [0,1]} \sup_{x \in \mathbb{R}} \left| \ddot{F}_t(x) \right| \leq C, \qquad \sup_{t \in [0,1]} \sup_{x,y \in \mathbb{R}, x \neq y} \frac{\left| \ddot{F}_t(x) - \ddot{F}_t(y) \right|}{|x-y|} \leq C.$$

It follows from Taylor's expansion that for any $x, y \in \mathbb{R}$, and $t \in [0,1]$,

$$\left| F_t(x+y) - F_t(x) - \dot{F}_t(x)y - \frac{1}{2} \ddot{F}_t(x)y^2 \right| \leq \frac{C}{2} |y|^3. \quad (19)$$

For any $1 \leq m \leq n$, taking $x = T_{m-1,\lambda}(\theta_n)$, $y = \frac{\lambda \bar{R}_m^{(\vartheta_m)}}{(1-\lambda)n} + \frac{R_m^{(\vartheta_m)}}{\sqrt{n}\hat{\sigma}}$ in (19), we obtain

$$\sum_{m=1}^{n} \mathbb{E} \left[ \left| F_{\frac{m}{n}}(T_{m,\lambda}(\theta_n)) - \Gamma(m, n, \theta_n) \right| \right]$$

$$\leq \frac{C_1}{2} \sum_{m=1}^{n} \mathbb{E} \left[ \left| \frac{\lambda \bar{R}_m^{(\vartheta_m)}}{(1-\lambda)n} \right|^2 + 2 \left| \frac{\lambda \bar{R}_m^{(\vartheta_m)}}{(1-\lambda)n} \right| \left| \frac{R_m^{(\vartheta_m)}}{\sqrt{n}\hat{\sigma}} \right| + \left| \frac{\lambda \bar{R}_m^{(\vartheta_m)}}{(1-\lambda)n} + \frac{R_m^{(\vartheta_m)}}{\sqrt{n}\hat{\sigma}} \right|^3 \right]$$

$$\leq \frac{C_1}{2} \left( \frac{2\sigma}{(1-\lambda)\sqrt{n}} + \frac{\lambda^2}{(1-\lambda)^2 n} + \frac{3\lambda\sigma}{(1-\lambda)n} \right) \leq \frac{C_1 \sigma}{(1-\lambda)\sqrt{n}},$$

The penultimate inequality is due to the uniform boundedness of $\{R_m^{(\vartheta_m)}\}$. Therefore, Equation (18) holds evidently. $\qquad \square$

**Lemma B.3.** *Define the family of functions* $\{L_{m,n}(x)\}_{m=1}^{n}$ *and* $\{\widehat{L}_{m,n}(x)\}_{m=1}^{n}$ *by*

$$L_{m,n}(x) = F_{\frac{m}{n}}(x) - \frac{\omega_n}{n}\left|\dot{F}_{\frac{m}{n}}(x)\right| + \frac{\sigma_0^2}{2n}\ddot{F}_{\frac{m}{n}}(x), \quad x \in \mathbb{R}, \tag{20}$$

$$\widehat{L}_{m,n}(x) = F_{\frac{m}{n}}(x) + \frac{\omega_n}{n}\left|\dot{F}_{\frac{m}{n}}(x)\right| + \frac{\sigma_0^2}{2n}\ddot{F}_{\frac{m}{n}}(x), \quad x \in \mathbb{R}, \tag{21}$$

*where* $\omega_n$ *and* $\sigma_0$ *is given in (12). Let* $\theta_n^*$ *be the strategy given in Theorem 4.1, then the followings hold.*

*(1) If* $\mathrm{sgn}(\dot{\varphi}(x)) = -\mathrm{sgn}(x)$ *for all* $x \in \mathbb{R}$*, then*

$$\sum_{m=1}^{n}\left|\mathbb{E}\left[F_{\frac{m}{n}}\left(T_{m,\lambda}(\theta_n^*)\right)\right] - \mathbb{E}\left[L_{m,n}\left(T_{m-1,\lambda}(\theta_n^*)\right)\right]\right| = O\left(\frac{\sigma}{(1-\lambda)\sqrt{n}}\right). \tag{22}$$

*(2) If* $\mathrm{sgn}(\dot{\varphi}(x)) = \mathrm{sgn}(x)$ *for all* $x \in \mathbb{R}$*, then*

$$\sum_{m=1}^{n}\left|\mathbb{E}\left[F_{\frac{m}{n}}\left(T_{m,\lambda}(\theta_n^*)\right)\right] - \mathbb{E}\left[\widehat{L}_{m,n}\left(T_{m-1,\lambda}(\theta_n^*)\right)\right]\right| = O\left(\frac{\sigma}{(1-\lambda)\sqrt{n}}\right). \tag{23}$$

*Proof.* We only give the proof of (1), the rest of the proofs are similar. For any $x \in \mathbb{R}$, $\mathrm{sgn}(\dot{\varphi}(x)) = -\mathrm{sgn}(x)$. It follows from (3) in Lemma B.1 and direct calculation that, for $1 \leq m \leq n$,

$$\mathbb{E}\left[\Gamma\left(m,n,\theta_n^*\right)\right]$$

$$=\mathbb{E}\left[F_{\frac{m}{n}}\left(T_{m-1,\lambda}(\theta_n^*)\right) + \dot{F}_{\frac{m}{n}}\left(T_{m-1,\lambda}(\theta_n^*)\right)\left(\frac{\lambda\bar{R}_m^{(\vartheta_m^*)}}{(1-\lambda)n} + \frac{R_m^{(\vartheta_m^*)}}{\sqrt{n}\hat{\sigma}}\right) + \frac{1}{2}\ddot{F}_{\frac{m}{n}}\left(T_{m-1,\lambda}(\theta_n^*)\right)\left(\frac{R_m^{(\vartheta_m^*)}}{\sqrt{n}\hat{\sigma}}\right)^2\right]$$

$$=\mathbb{E}\left[F_{\frac{m}{n}}\left(T_{m-1,\lambda}(\theta_n^*)\right) + I_{\{\vartheta_m^*=1\}}\dot{F}_{\frac{m}{n}}\left(T_{m-1,\lambda}(\theta_n^*)\right)\left(\frac{\lambda\bar{R}_m^{(1)}}{(1-\lambda)n} + \frac{R_m^{(1)}}{\sqrt{n}\hat{\sigma}}\right)\right.$$

$$\left.+ I_{\{\vartheta_m^*=0\}}\dot{F}_{\frac{m}{n}}\left(T_{m-1,\lambda}(\theta_n^*)\right)\left(\frac{\lambda\bar{R}_m^{(0)}}{(1-\lambda)n} + \frac{R_m^{(0)}}{\sqrt{n}\hat{\sigma}}\right) + \ddot{F}_{\frac{m}{n}}\left(T_{m-1,\lambda}(\theta_n^*)\right)\frac{(R_m^{(1)})^2}{2n\hat{\sigma}^2}\right]$$

$$=\mathbb{E}\left[L_{m,n}\left(T_{m-1,\lambda}(\theta_n^*)\right)\right],$$

then, according to (1) of Lemma B.1 and Equation (18), we have

$$\sum_{m=1}^{n}\left|\mathbb{E}\left[F_{\frac{m}{n}}\left(T_{m,\lambda}(\theta_n^*)\right)\right] - \mathbb{E}\left[L_{m,n}\left(T_{m-1,\lambda}(\theta_n^*)\right)\right]\right|$$

$$= \sum_{m=1}^{n}\left|\mathbb{E}\left[F_{\frac{m}{n}}\left(T_{m,\lambda}(\theta_n^*)\right)\right] - \mathbb{E}\left[\Gamma\left(m,n,\theta_n^*\right)\right]\right| \tag{24}$$

$$\leq \frac{C_1\sigma}{(1-\lambda)\sqrt{n}}.$$

$\square$

**Lemma B.4.** *With the assumptions and notations in Lemma B.3, the followings hold.*

*(1) If* $\mathrm{sgn}(\dot{\varphi}(x)) = -\mathrm{sgn}(x)$ *for all* $x \in \mathbb{R}$*, then*

$$\sum_{m=1}^{n}\left|\sup_{\theta_n\in\Theta}\mathbb{E}\left[F_{\frac{m}{n}}\left(T_{m,\lambda}(\theta_n)\right)\right] - \sup_{\theta_n\in\Theta}\mathbb{E}\left[L_{m,n}\left(T_{m-1,\lambda}(\theta_n)\right)\right]\right| = O\left(\frac{\sigma}{(1-\lambda)\sqrt{n}}\right). \tag{25}$$

*(2) If* $\mathrm{sgn}(\dot{\varphi}(x)) = \mathrm{sgn}(x)$ *for all* $x \in \mathbb{R}$*, then*

$$\sum_{m=1}^{n}\left|\sup_{\theta_n\in\Theta}\mathbb{E}\left[F_{\frac{m}{n}}\left(T_{m,\lambda}(\theta_n)\right)\right] - \sup_{\theta_n\in\Theta}\mathbb{E}\left[\widehat{L}_{m,n}\left(T_{m-1,\lambda}(\theta_n)\right)\right]\right| = O\left(\frac{\sigma}{(1-\lambda)\sqrt{n}}\right). \tag{26}$$

*Proof.* We only give the proof of (1), the rest of the proofs are similar. For any $x \in \mathbb{R}$, $\text{sgn}(\dot{\varphi}(x)) = -\text{sgn}(x)$. It follows from (3) in Lemma B.1 and direct calculation that, for $1 \leq m \leq n$,

$$\sup_{\theta_n \in \Theta} \mathbb{E}\left[\Gamma(m, n, \theta_n)\right]$$

$$= \sup_{\theta_n \in \Theta} \mathbb{E}\left[F_{\frac{m}{n}}(T_{m-1,\lambda}(\theta_n)) + \dot{F}_{\frac{m}{n}}(T_{m-1,\lambda}(\theta_n))\left(\frac{\lambda \bar{R}_m^{(\vartheta_m)}}{(1-\lambda)n} + \frac{R_m^{(\vartheta_m)}}{\sqrt{n}\hat{\sigma}}\right) + \frac{1}{2}\ddot{F}_{\frac{m}{n}}(T_{m-1,\lambda}(\theta_n))\left(\frac{R_m^{(\vartheta_m)}}{\sqrt{n}\hat{\sigma}}\right)^2\right]$$

$$= \sup_{\theta_n \in \Theta} \mathbb{E}\left[F_{\frac{m}{n}}(T_{m-1,\lambda}(\theta_n)) - \left(\dot{F}_{\frac{m}{n}}(T_{m-1,\lambda}(\theta_n))\right)^-\left(\frac{\lambda \bar{R}_m^{(1)}}{(1-\lambda)n} + \frac{R_m^{(1)}}{\sqrt{n}\hat{\sigma}}\right)\right.$$

$$\left. + \left(\dot{F}_{\frac{m}{n}}(T_{m-1,\lambda}(\theta_n))\right)^+\left(\frac{\lambda \bar{R}_m^{(0)}}{(1-\lambda)n} + \frac{R_m^{(0)}}{\sqrt{n}\hat{\sigma}}\right) + \ddot{F}_{\frac{m}{n}}(T_{m-1,\lambda}(\theta_n))\frac{(R_m^{(1)})^2}{2n\hat{\sigma}^2}\right]$$

$$= \sup_{\theta_n \in \Theta} \mathbb{E}\left[L_{m,n}(T_{m-1,\lambda}(\theta_n))\right],$$

then, according to (1) of Lemma B.1 and Equation (18), we have

$$\sum_{m=1}^{n}\left|\sup_{\theta_n \in \Theta} \mathbb{E}\left[F_{\frac{m}{n}}(T_{m,\lambda}(\theta_n))\right] - \sup_{\theta_n \in \Theta} \mathbb{E}\left[L_{m,n}(T_{m-1,\lambda}(\theta_n^*))\right]\right|$$

$$= \sum_{m=1}^{n}\left|\sup_{\theta_n \in \Theta} \mathbb{E}\left[F_{\frac{m}{n}}(T_{m,\lambda}(\theta_n))\right] - \sup_{\theta_n \in \Theta} \mathbb{E}\left[\Gamma(m, n, \theta_n)\right]\right| \tag{27}$$

$$\leq \frac{C_1 \sigma}{(1-\lambda)\sqrt{n}}.$$

$\square$

Having presented the aforementioned lemma, we now proceed to prove Theorem 4.1.

*Proof.* **(Proof of Theorem 4.1.)** Let $\varphi \in C(\bar{\mathbb{R}})$ be an even function. The result is clear if $\varphi$ is globally constant. Thus, we assume that $\varphi$ is not a constant function. We only give the proof for the case that $\varphi$ is decreasing on $(0, \infty)$, when $\varphi$ is increasing on $(0, \infty)$ it can be proved similarly. Assume that $\varphi$ is decreasing on $(0, \infty)$. For any $h > 0$, define the function $\varphi_h$ by

$$\varphi_h(x) = \int_{-\infty}^{\infty} \frac{1}{\sqrt{2\pi}} \varphi(x + hy) e^{-\frac{y^2}{2}} dy.$$

By the Approximation Lemma, we have that

$$\lim_{h \to 0} \sup_{x \in \mathbb{R}} |\varphi(x) - \varphi_h(x)| = 0. \tag{28}$$

It follows from direct calculation that

$$\varphi_h(x) = \int_{-\infty}^{\infty} \frac{1}{\sqrt{2\pi}} \varphi(x + hy) e^{-\frac{y^2}{2}} dy = \int_{-\infty}^{\infty} \frac{1}{\sqrt{2\pi}} \varphi(-x + hy) e^{-\frac{y^2}{2}} dy = \varphi_h(-x).$$

Thus, $\varphi_h$ is symmetric with centre $c$. In addition, we have

$$\dot{\varphi}_h(x) = \int_{-\infty}^{\infty} \frac{1}{\sqrt{2\pi}h^3} \varphi(x + y) y e^{-\frac{y^2}{2h^2}} dy$$

$$= \int_{0}^{\infty} \frac{1}{\sqrt{2\pi}h^3} \varphi(y + x) y e^{-\frac{y^2}{2h^2}} dy + \int_{-\infty}^{0} \frac{1}{\sqrt{2\pi}h^3} \varphi(y + x) y e^{-\frac{y^2}{2h^2}} dy$$

$$= \int_{0}^{\infty} \frac{1}{\sqrt{2\pi}h^3} (\varphi(y + x) - \varphi(y - x)) y e^{-\frac{y^2}{2h^2}} dy.$$

Since $\varphi$ is decreasing on $(0, \infty)$, it follows that

$$\mathrm{sgn}\left(\dot{\varphi}_h(x)\right) = -\mathrm{sgn}(x).$$

In the remaining proof of this theorem, we continue to use $\{F_t(x)\}_{t \in [0,1]}$ to denote the functions defined in Equation (13) with $\varphi_h$ in place of $\varphi$ and $\delta = \omega_n$, $\beta = \sigma_0$ there. Let $\{L_{m,n}(x)\}_{m=1}^n$ be functions defined in Equation (20) with $\{F_t(x)\}_{t \in [0,1]}$ here. Let $\eta_n \sim \mathcal{B}(\omega_n, \sigma_0)$, by direct calculation we obtain

$$
\begin{aligned}
\mathbb{E}\left[\varphi_h\left(T_{n,\lambda}(\theta_n^*)\right)\right] - \mathbb{E}\left[\varphi_h\left(\eta_n\right)\right] =& \mathbb{E}\left[F_1\left(T_{n,\lambda}(\theta_n^*)\right)\right] - F_0(0) \\
=& \mathbb{E}\left[F_1\left(T_{n,\lambda}(\theta_n^*)\right)\right] - \mathbb{E}\left[F_{\frac{n-1}{n}}\left(T_{n-1,\lambda}(\theta_n^*)\right)\right] + \ldots + \mathbb{E}\left[F_{\frac{m}{n}}\left(T_{m,\lambda}(\theta_n^*)\right)\right] \\
& - \mathbb{E}\left[F_{\frac{m-1}{n}}\left(T_{m-1,\lambda}(\theta_n^*)\right)\right] + \ldots + \mathbb{E}\left[F_{\frac{1}{n}}\left(T_{1,\lambda}(\theta_n^*)\right)\right] - F_0\left(T_{0,\lambda}(\theta_n^*)\right) \\
=& \sum_{m=1}^n \left\{\mathbb{E}\left[F_{\frac{m}{n}}\left(T_{m,\lambda}(\theta_n^*)\right)\right] - \mathbb{E}\left[F_{\frac{m-1}{n}}\left(T_{m-1,\lambda}(\theta_n^*)\right)\right]\right\} \\
=& \sum_{m=1}^n \left\{\mathbb{E}\left[F_{\frac{m}{n}}\left(T_{m,\lambda}(\theta_n^*)\right)\right] - \mathbb{E}\left[L_{m,n}\left(T_{m-1,\lambda}(\theta_n^*)\right)\right]\right\} \\
& + \sum_{m=1}^n \left\{\mathbb{E}\left[L_{m,n}\left(T_{m-1,\lambda}(\theta_n^*)\right)\right] - \mathbb{E}\left[F_{\frac{m-1}{n}}\left(T_{m-1,\lambda}(\theta_n^*)\right)\right]\right\} \\
=:& I_{1n} + I_{2n}.
\end{aligned}
$$

According to Lemma B.3 and (4) in Lemma B.1, we can infer

$$|I_{1n}| + |I_{2n}| \leq K'\left(\frac{|\omega_n|\sigma}{n} + \frac{\sigma}{(1-\lambda)\sqrt{n}}\right).$$

Which implies that

$$\lim_{h \to 0}\left|\mathbb{E}\left[\varphi_h\left(T_{n,\lambda}(\theta_n^*)\right)\right] - \mathbb{E}\left[\varphi_h\left(\eta_n\right)\right]\right| = O\left(\frac{\lambda|\mu|\sigma}{(1-\lambda)n} + \frac{\sigma}{(1-\lambda)\sqrt{n}}\right). \tag{29}$$

Putting together Equation (28) and (29), we have

$$
\begin{aligned}
\lim_{n \to \infty}\left|\mathbb{E}\left[\varphi\left(T_{n,\lambda}(\theta_n^*)\right)\right] - \mathbb{E}\left[\varphi\left(\eta_n\right)\right]\right| \leq & \lim_{h \to 0}\lim_{n \to \infty}\left|\mathbb{E}\left[\varphi\left(T_{n,\lambda}(\theta_n^*)\right)\right] - \mathbb{E}\left[\varphi_h\left(T_{n,\lambda}(\theta_n^*)\right)\right]\right| \\
& + \lim_{h \to 0}\lim_{n \to \infty}\left|\mathbb{E}\left[\varphi_h\left(T_{n,\lambda}(\theta_n^*)\right)\right] - \mathbb{E}\left[\varphi_h\left(\eta_n\right)\right]\right| \\
& + \lim_{h \to 0}\left|\mathbb{E}\left[\varphi_h\left(\eta_n\right)\right] - \mathbb{E}\left[\varphi\left(\eta_n\right)\right]\right| \\
= & 0.
\end{aligned}
$$

where $\eta_n \sim \mathcal{B}(\omega_n, \sigma_0)$. Then we complete the proof of Theorem 4.1. $\qquad \square$

Theorem 4.2 is a corollary directly from Theorem 4.1. We still give the Proof of Theorem 4.2 here.

*Proof.* **(Proof of Theorem 4.2.)** Let $\varphi \in C(\overline{\mathbb{R}})$ be an even function and monotonic on $(0, \infty)$. We first prove that

$$\lim_{n \to \infty}\left|\mathbb{E}\left[\varphi\left(T_{n,\lambda}(\theta_n^*)\right)\right] - E\left[\varphi\left(\eta_n\right)\right]\right| = 0$$

where $\eta_n \sim \mathcal{B}(\omega_n, \sigma_0)$ is a spike distribution with the parameter $\omega_n$, $\sigma_0$ given in Equation (12).

The result is clear if $\varphi$ is globally constant. Thus, we assume that $\varphi$ is not a constant function. We only give the proof for the case that $\varphi$ is decreasing on $(0, \infty)$, when $\varphi$ is increasing on $(0, \infty)$ it can be proved similarly.

Assume that $\varphi$ is decreasing on $(0, \infty)$. For any $h > 0$, define the function $\varphi_h$ by

$$\varphi_h(x) = \int_{-\infty}^{\infty} \frac{1}{\sqrt{2\pi}}\varphi(x + hy)\mathrm{e}^{-\frac{y^2}{2}} \, \mathrm{d}y.$$

By the Approximation Lemma, we have that

$$\lim_{h\to 0}\sup_{x\in\mathbb{R}}|\varphi(x)-\varphi_h(x)|=0.$$

By the proof of Theorem 4.1, we also have $\varphi_h$ is an even function, an when $\varphi$ is decreasing on $(0,\infty)$, it follows that

$$\mathrm{sgn}\left(\dot\varphi_h(x)\right)=-\mathrm{sgn}(x).$$

In the remaining proof of this theorem, we continue to use $\{F_t(x)\}_{t\in[0,1]}$ to denote the functions defined in (13) with $\varphi_h$ in place of $\varphi$ and $\delta=\omega_n$, $\beta=\sigma_0$ there. Let $\{L_{m,n}(x)\}_{m=1}^n$ be functions defined in (22) with $\{F_t(x)\}_{t\in[0,1]}$ here. Let $\eta_n\sim\mathcal{B}\left(\omega_n,\sigma_0\right)$ be a spike distribution, by direct calculation we obtain

$$
\begin{aligned}
\mathbb{E}\left[\varphi_h\left(T_{n,\lambda}(\theta_n^*)\right)\right]-\mathbb{E}\left[\varphi_h\left(\eta_n\right)\right]=&\mathbb{E}\left[F_1\left(T_{n,\lambda}(\theta_n^*)\right)\right]-F_0(0)\\
=&\mathbb{E}\left[F_1\left(T_{n,\lambda}(\theta_n^*)\right)\right]-\mathbb{E}\left[F_{\frac{n-1}{n}}\left(T_{n-1,\lambda}(\theta_n^*)\right)\right]+\ldots+\mathbb{E}\left[F_{\frac{m}{n}}\left(T_{m,\lambda}(\theta_n^*)\right)\right]\\
&-\mathbb{E}\left[F_{\frac{m-1}{n}}\left(T_{m-1,\lambda}(\theta_n^*)\right)\right]+\ldots+\mathbb{E}\left[F_{\frac{1}{n}}\left(T_{1,\lambda}(\theta_n^*)\right)\right]-F_0\left(T_{0,\lambda}(\theta_n^*)\right)\\
=&\sum_{m=1}^n\left\{\mathbb{E}\left[F_{\frac{m}{n}}\left(T_{m,\lambda}(\theta_n^*)\right)\right]-\mathbb{E}\left[F_{\frac{m-1}{n}}\left(T_{m-1,\lambda}(\theta_n^*)\right)\right]\right\}\\
=&\sum_{m=1}^n\left\{\mathbb{E}\left[F_{\frac{m}{n}}\left(T_{m,\lambda}(\theta_n^*)\right)\right]-\mathbb{E}\left[L_{m,n}\left(T_{m-1,\lambda}(\theta_n^*)\right)\right]\right\}\\
&+\sum_{m=1}^n\left\{\mathbb{E}\left[L_{m,n}\left(T_{m-1,\lambda}(\theta_n^*)\right)\right]-\mathbb{E}\left[F_{\frac{m-1}{n}}\left(T_{m-1,\lambda}(\theta_n^*)\right)\right]\right\}\\
=&:I_{1n}+I_{2n}.
\end{aligned}
$$

According to Lemma B.3 and (4) in Lemma B.1, we can infer

$$\lim_{h\to 0}\lim_{n\to\infty}\left|\mathbb{E}\left[\varphi_h\left(T_{n,\lambda}(\theta_n^*)\right)\right]-\mathbb{E}\left[\varphi_h\left(\eta_n\right)\right]\right|=0.$$

Putting together Equation (28) and (29), we have

$$
\begin{aligned}
\lim_{n\to\infty}\left|\mathbb{E}\left[\varphi\left(T_{n,\lambda}(\theta_n^*)\right)\right]-\mathbb{E}\left[\varphi\left(\eta_n\right)\right]\right|\leq&\lim_{h\to 0}\lim_{n\to\infty}\left|\mathbb{E}\left[\varphi\left(T_{n,\lambda}(\theta_n^*)\right)\right]-\mathbb{E}\left[\varphi_h\left(T_{n,\lambda}(\theta_n^*)\right)\right]\right|\\
&+\lim_{h\to 0}\lim_{n\to\infty}\left|\mathbb{E}\left[\varphi_h\left(T_{n,\lambda}(\theta_n^*)\right)\right]-\mathbb{E}\left[\varphi_h\left(\eta_n\right)\right]\right|\\
&+\lim_{h\to 0}\left|\mathbb{E}\left[\varphi_h\left(\eta_n\right)\right]-\mathbb{E}\left[\varphi\left(\eta_n\right)\right]\right|\\
=&\,0.
\end{aligned}
$$

With the standard approximation arguments, we have for any $a\in\mathbb{R}$, we have

$$\lim_{n\to\infty}\left\{P\left(|T_{n,\lambda}(\theta_n^*)|\leq a\right)-\left[\Phi\left(\frac{\omega_n-a}{\sigma_0}\right)-e^{\frac{2\omega_n a}{\sigma_0^2}}\Phi\left(-\frac{\omega_n+a}{\sigma_0}\right)\right]\right\}=0.$$

Next, when $\varphi$ is decreasing on $(0,\infty)$, by Lemma B.4 (1), with the similar arguments as above, we have

$$\lim_{n\to\infty}\left|\sup_{\theta_n\in\Theta}\mathbb{E}\left[\varphi_h\left(T_{n,\lambda}(\theta_n)\right)\right]-\mathbb{E}\left[\varphi_h\left(\eta_n\right)\right]\right|=0$$

and for any $a\in\mathbb{R}$, we have

$$\lim_{n\to\infty}\left\{\sup_{\theta_n\in\Theta}P\left(|T_{n,\lambda}(\theta_n)|\leq a\right)-\left[\Phi\left(\frac{\omega_n-a}{\sigma_0}\right)-e^{\frac{2\omega_n a}{\sigma_0^2}}\Phi\left(-\frac{\omega_n+a}{\sigma_0}\right)\right]\right\}=0.$$

Then, we complete the proof of Theorem 4.2. $\qquad\square$

To ensure the completeness of the proof, we now present the derivation of Lemma 2.1 within our framework, independently of Theorem 3.3 in Chen et al. (2022), by considering the case where $\lambda = 0.5$ and assuming an oracle scenario in which the counterfactual outcomes $Y_i^{(1)}$ and $Y_i^{(0)}$ for each subject can be observed simultaneously.

*Proof.* **(Proof of Lemma 2.1.)** First, when $\lambda = 0.5$ and the counterfactual outcomes $Y_i^{(1)}$ and $Y_i^{(0)}$ for each subject can be observed simultaneously, it is clear from Theorem 4.1 that we can derive

$$E\left[|\varphi\left(T_n(\theta_n^*)\right) - \varphi\left(\eta_n\right)|\right] = O\left(\frac{\sigma}{\sqrt{n}}\right).$$

With the standard approximation arguments, we have for any $\alpha \in [0, 1]$, we have

$$\lim_{n\to\infty} \left\{ P\left(|T_n(\theta_n^*)| > z_{1-\alpha/2}\big|\mathcal{H}_1\right) - \left[\Phi\left(\frac{\omega_n' - z_{1-\alpha/2}}{\sigma_0}\right) + e^{\frac{2\omega_n' z_{1-\alpha/2}}{\sigma_0^2}} \Phi\left(-\frac{\omega_n' + z_{1-\alpha/2}}{\sigma_0}\right)\right] \right\} = 0,$$

where $\omega_n' = \mu + \sqrt{n}\mu/\sigma$ and $z_\alpha$ denotes the $\alpha$th quantile of a standard normal distribution. Next, when $\varphi$ is decreasing on $(0, \infty)$, by Lemma B.4 (1), with the similar arguments as above, we have

$$\lim_{n\to\infty} \left| \sup_{\theta_n \in \Theta} E\left[\varphi\left(T_n(\theta_n)\right)\right] - E\left[\varphi\left(\eta_n\right)\right] \right| = 0,$$

and for any $\alpha \in [0, 1]$, we have

$$\lim_{n\to\infty} \left\{ \sup_{\theta_n \in \Theta} P\left(|T_n(\theta_n)| > z_{1-\alpha/2}\big|\mathcal{H}_1\right) - \left[\Phi\left(\frac{\omega_n' - z_{1-\alpha/2}}{\sigma_0}\right) + e^{\frac{2\omega_n' z_{1-\alpha/2}}{\sigma_0^2}} \Phi\left(-\frac{\omega_n' + z_{1-\alpha/2}}{\sigma_0}\right)\right] \right\} = 0.$$

Therefore, it is clear that we can obtain

$$\lim_{n\to\infty} \mathbb{P}\left(|T_n(\theta_n^*)| > z_{1-\alpha/2}\big|\mathcal{H}_1\right) = \lim_{n\to\infty} \sup_{\theta_n \in \Theta} \mathbb{P}\left(|T_n(\theta_n)| > z_{1-\alpha/2}\big|\mathcal{H}_1\right).$$

Similarly, when $\varphi$ is increasing on $(0, \infty)$, the above conclusion still holds by Lemma B.4 and Theorem 4.1. Then, we complete the proof of Lemma 2.1. $\qquad\square$

