# OpenReview forum: "Strategic A/B testing via Maximum Probability-driven Two-armed Bandit"
_ICML.cc/2025/Conference — ICML 2025 poster_

### Official Review · Reviewer_e8M1 · 2025-03-12

**Overall Recommendation:** 1

**Summary:**

This paper builds on Strategic Two-Sample Test via the Two-Armed Bandit Process to enhance the detection of small average treatment effects. It proposes a more powerful one-sided two-sample test by adjusting the balance between the mean and volatility terms, yielding a statistic that is more concentrated under the null and less so under the alternative. The framework is adapted to the Rubin Causal Model (RCM), where only one potential outcome per subject is observed, with a doubly robust estimator used for causal effect imputation. To address sensitivity to sample ordering, the authors incorporate meta-analysis by repeatedly reordering samples and recalculating p-values. Theoretically, they show that as $n$ approaches infinity, the asymptotic distribution converges to a spike, ensuring valid inference.

**Claims And Evidence:**

The validity of the proposed statistic is questionable. For instance, in Equation (2), classical statistical inference typically expresses the mean term as the sample average of $R_i^{(\vartheta_i)}$ for $i = 1,2,\dots,n$. However, the authors instead define it as $\bar{R}n^{(\vartheta_i)} = \sum{j=1}^n R_j^{(\vartheta_i)} / n$, raising a fundamental issue: what is the meaning of $R_j^{(\vartheta_i)}$ when $i \neq j$? Since this formulation underpins the entire paper, its incorrect mathematical structure casts doubt on the validity of the conclusions, theory, and experiments. The authors should carefully reexamine their theoretical derivations, algorithmic details, and experimental code.

**Essential References Not Discussed:**

I mention only one important paper, Strategic Two-Sample Test via the Two-Armed Bandit Process. While this paper does not cite the foundational work, much of its content serves as a technical extension of it.

**Experimental Designs Or Analyses:**

Yes.

**Methods And Evaluation Criteria:**

Yes.

**Other Comments Or Suggestions:**

No.

**Other Strengths And Weaknesses:**

Strengths: The weighting and permutation tricks is beneficial for finite-sample performance in statistical inference.

**Questions For Authors:**

No.

**Relation To Broader Scientific Literature:**

This paper primarily addresses the problem of one-sided two-sample testing, with a particular focus on paired two-sample testing. Compared to previous works, such as Strategic Two-Sample Test via the Two-Armed Bandit Process, which establish a more comprehensive theoretical framework—including proofs of the strategic central limit theorem (CLT) and other asymptotic properties—this paper places greater emphasis on practical implementation and the technical challenges that arise in finite samples.

Despite repeatedly highlighting its connection to the two-armed bandit, the constructed bandit framework assigns rewards to the two arms as exact opposites, effectively reducing it to a one-armed bandit. Moreover, in traditional two-sample testing, data from both populations are fully observed, eliminating the partial observation challenge that is central to bandit problems. As a result, this paper primarily addresses two-sample testing and has only a limited connection to the bandit literature.

**Theoretical Claims:**

Yes. The proofs for the theorems on the asymptotic properties of PWTAB are standard and rely on the strategic CLT. However, as noted in the Claims and Evidence section, the mathematical formulation of the proposed statistics seems incorrect, casting doubt on the validity of the theoretical results.

---

> ### Author Rebuttal · Authors · 2025-03-31
>
> We sincerely appreciate the reviewer’s thoughtful feedback and valuable suggestions. We also sincerely apologize for any difficulties or confusion arising from insufficient clarity in the presentation of some fundamental equations and references in the paper. Below, we will discuss the issues you have raised：
>
> **1. Effectiveness of the Proposed Statistic**
>
> - **Notation Clarification:**
>
> We define  $\bar{R}_n^{(\vartheta_i)}=\bar{R}_n^{(1)}\mathbb{I}(\vartheta_i=1)+\bar{R}_n^{(0)}\mathbb{I}(\vartheta_i=0)$, where $\bar{R}_n^{(1)}=\sum\_{i=1}^nR_i^{(1)}/n=-\bar{R}^{(0)}_n$ and $\mathbb{I}(\cdot)$ denotes the indicator function. Since our paper does not introduce or define the notation $R_j^{(\vartheta_i)}$ for $i \neq j$, we will explicitly define $\bar{R}_n^{(\vartheta_i)}$ as stated above, immediately following Equation (2).
>
> - **Intuitive Explanation and Theoretical Perspective:**
>
> Using $\sum\_{i=1}^n\bar{R}\_n^{(\vartheta_i)}/n$ instead of $\sum\_{i=1}^n R_i^{(\vartheta_i)}/n$ reduces the variance of the mean term while preserving the asymptotic properties of the statistic. This leads to a faster and more stable convergence, as also verified experimentally. Replacing $R_i^{(\vartheta_i)}$ with $\bar{R}_n^{(1)}\mathbb{I}(\vartheta_i=1)+\bar{R}_n^{(0)}\mathbb{I}(\vartheta_i=0)$ represents a key methodological improvement.
>
> **2. Motivation for Linking to the Two-Armed Bandit Framework**
>
> - **Differences in Research Objectives:**
>
> The classical two-armed bandit model focuses on maximizing the average reward by balancing exploration and exploitation. In contrast, our model does not seek to identify the arm with the highest return. Instead, it aims to maximize the target probability through collaborative arm selection. Within the hypothesis testing framework, this maximized target probability corresponds to the tail probability $\mathbb{P}(|T\_{n, \lambda}(\theta_n)|>z_{1-\alpha/2}|\mathcal H_1)$.
>
> - **Clarification in Writing:**
>
> Given the distinct objectives of our proposed model compared to the classical framework, we appreciate the opportunity to improve clarity in our manuscript. To this end, we will explicitly define our proposed model following the second paragraph of Section 2.2, thereby clearly distinguishing it from the classical model.
>
> **3. Missing References**
>
> We sincerely apologize for the unintentional omission of the reference to “Z. Chen et al., Strategic Two-Sample Test via the Two-Armed Bandit Process” during the drafting process. This reference has been included in the revised manuscript along with a detailed comparison:
>
> - **Research Background:**
>
> Chen et al. addressed the one-sided two-sample testing problem under independent, batch-wise paired observations. While valuable, their approach is not designed to address hypothesis testing in more complex frameworks, such as causal inference scenarios involving missing data or confounding variables. In contrast, our proposed method incorporates advanced techniques, enabling its application to hypothesis testing within causal inference frameworks. This adaptability enhances its relevance to real-world research contexts.
>
> - **Theoretical Contributions:**
>
> Chen et al.’s study primarily focuses on the asymptotic properties of the test statistic, without investigating its behavior in finite samples. This limitation can result in inflated Type I errors in small-sample settings. We effectively control the Type I error under finite samples by modifying the mean term as the mean of $\bar{R}\_n^{(\vartheta_i)}$ and introducing a weighting factor $\lambda$. Their result represents a special case of our framework when $\lambda=0.5$.
>
> - **Algorithm Robustness:**
>
> Chen et al. compute the $p$-value from a single ordered sample sequence, resulting in unstable statistical power in both simulations and real-world experiments. To enhance robustness, we introduce a permutation-based meta-analysis, recalculating the $p$-value across multiple sample reorderings. This improvement significantly strengthens the algorithm’s reliability and practical utility.
>
> In summary, while our work draws inspiration from “Strategic Two-Sample Test via the Two-Armed Bandit Process” in its use of bandit strategies for hypothesis testing, our study offers more generalized insights across research frameworks, theoretical advancements, and technical implementations. We hope this clarification underscores the distinct contributions of our work while acknowledging the foundational influence of Chen et al.’s research.
>
> We sincerely appreciate the reviewer’s keen observation, which has enabled us to refine the manuscript and better contextualize our contributions within the existing literature.

---

### Official Review · Reviewer_QGjr · 2025-03-12

**Overall Recommendation:** 4

**Summary:**

**Problem:**

This work aims to address the limitations of traditional A/B testing in detecting minor treatment effects.
The key challenges are: (i) data distributions between the treatment and control groups may differ due to confounding effects, (ii) even when distributions are balanced, measured outcomes can still exhibit high variance, and (iii) test statistics rely on the normality assumption of the central limit theorem, which may not always hold.

**Methods Used:**

To this end, this work proposes a novel statistical testing framework that: (i) relaxes the normality assumption by leveraging bandit-inspired distributions, which is drawn from the the prior results of strategy-central limit theorem (Chen et al., 2022) [1], (ii) introduces a weighted test statistic to control Type I error. (iii) employs a doubly robust method to obtain unbiased, low-variance causal estimates, and (iv) utilizes a permutation test to enhance statistical power.


**Results:**

On empirical evaluations, the authors compare their methods, i.e., Permuted WTAB and WTAB,  with existing methods (i.e., z-DML, CUPED, and DIM) on both synthetic data and real-world ridge sharing data.

On synthetic data, the authors show that their Permuted WTAB achieves the highest statistical powers compared to other methods while maintaining a similar Type I error rate.

Similarly, on real-world ridge sharing A/B testing datasets, they shows their Permuted WTAB consistently achieves lower p-values compared to CUPED.


**Overall — Main contributions, novelty and impact:**

1. This work **introduces the first hypothesis testing method for estimating causal effects, which breaks the assumption that random variables are normally distributed but instead Bandit distributed**, significantly improving the statistical power. **Its efficacy is further improved by weighted version and doubly-robust estimation.**
2. The proposed method is **novel** and **theoretically grounded**.
3. The proposed method has **major business implications**, particularly in optimizing A/B tests for small treatment effects, which could drive more profitable marketing strategies.

**Area Of Improvement:**

However, given the current empirical results, **additional evaluations are needed** to clarify whether this approach is indeed better than other existing. Moreover, their **current writing and presentation needs improvement**, which I will describe in later sections.


**Ref:**

[1] Chen, Zengjing, Shui Feng, and Guodong Zhang. "Strategy-driven limit theorems associated bandit problems." arXiv preprint arXiv:2204.04442 (2022). https://arxiv.org/pdf/2204.04442






=================================================================================================================================================


**Update After Rebuttal**

1. The authors have provided a clearer statement on the contribution of their work, particularly in how they leverage the strategic central limit theorem framework to relax the exchangeability assumption, thereby improving Type I error control while enhancing statistical power.

2. The authors have addressed the typo and the missing legend in Figure 2. They are also aware of areas where the presentation could be improved and have made an effort to address them.

3. They have added a discussion on how to select the regularization parameter λ in a data-driven manner and explained why the stacking approach did not perform as expected, along with suggestions for how to address this issue.

4. The authors have stated their intention to include the proof of Lemma 2.1.

An additional experiment on a real-world dataset has been included. The authors also explained the rationale for generating synthetic data from real-world sources, which is reasonable.

Given the above, I believe the authors have made a substantial effort to address the key concerns raised.

While we are unable to see the full revision due to the constraints of the rebuttal phase, I believe that the proposed revisions and clarifications significantly strengthen the work.

-- **I think the merits of the paper outweigh the remaining minor concerns (such as presentation)**, and **I therefore maintain my original recommendation of accept.**

-- **However**, if the committee feels it is preferable for the authors to take more time and submit a fully revised version in a future round, I would also support that direction.

=================================================================================================================================================

**Claims And Evidence:**

Their weighted version of the two-arm bandit with a permutation test effectively demonstrates control over Type I error while achieving higher statistical power than both state-of-the-art methods and the standard mean estimator on synthetic datasets.

However, in real-world datasets, they only present variance reduction and p-value results without providing analyses on Type I error control or statistical power.

**Essential References Not Discussed:**

The key paper to derive their results are provided, which is [1]

Ref:

[1] Chen, Zengjing, Shui Feng, and Guodong Zhang. "Strategy-driven limit theorems associated bandit problems." arXiv preprint arXiv:2204.04442 (2022). https://arxiv.org/pdf/2204.04442

**Experimental Designs Or Analyses:**

Their experimental designs and analyses are valid; however, additional evaluations on Type I error control and statistical power analysis for real-world datasets are needed to ensure comprehensive assessment.

**Methods And Evaluation Criteria:**

1. The authors appropriately utilize both synthetic and real-world datasets to evaluate their statistical testing methods, ensuring a comprehensive assessment of their approach across controlled and practical settings.

2. The authors selected both the baseline model, i.e., the mean average estimator, and state-of-the-art methods, such as DML and CUPED, to ensure a fair comparison.

3. As mentioned earlier, they also need evaluations on type 1 error and statistical power analysis for real-world dataset.

**Other Comments Or Suggestions:**

1. The authors make frequent use of the term exchangeability, but its precise meaning remains ambiguous. For instance, exchangeability might refer to data-level exchangeability, meaning whether the data are generated i.i.d. Alternatively, it could pertain to the exchangeability of treatment assignment given the observed data. In the context of the strategy central limit theorem, I interpret exchangeability as referring to the rewards derived from the sequence of arm choices. Clarifying this distinction would enhance the paper’s rigor and readability.

2. On line 73, it would be better to first introduce the full term Two-Arm Bandit before using its acronym.

3. Line 193: mean --> main

4. In Figure 2(b), there is no legend to describe what each curve means. Please include them.

5. In section 3.1, it would be better to firstly introduce what Theorem 4.1 is and then use it.

6. In your lemma 2.1,  please explicitly define $H_1$ as the alternative hypothesis before using it.

7.  In line 410, the paragraph titled "Another simulations" should be more clearly described. Please rephrase it to indicate that it presents results on an ML-based method or another relevant categorization for better clarity.

8. In general, the authors assume that readers are already familiar with the intent of each section and proceed without sufficient introduction. It would be beneficial to include brief overviews or contextual transitions at the beginning of each section to improve clarity and guide the reader through the flow of the paper.

**Other Strengths And Weaknesses:**

This work cleverly integrates strategy-based statistical testing, which challenges the normality assumption, with existing approaches such as the permutation test and the doubly-robust estimator. While the proof is largely adapted from prior work with slight modifications, its implications are significant in accurately detecting minor treatment effects.

**Questions For Authors:**

1. I may be wrong if I am not very scrutinizing. However, I am unsure if you have the proof for Lemma 2.1?

2. In line 258, you mention that the ensemble method, specifically stacking, improves efficacy. However, in your experiments on synthetic datasets (Figure 6), stacking is not consistently the best method and, in some cases (e.g., the bottom-left panel for function III), performs worse than the compared methods. Could you provide an intuition or explanation for why stacking underperforms in certain scenarios?

**Relation To Broader Scientific Literature:**

Unlike traditional normality-based hypothesis tests, this work introduces a Bandit-distributed framework, providing an alternative to standard A/B testing. The incorporation of weighted test statistics, doubly robust estimation, and permutation testing further strengthens treatment effect estimation.

**Theoretical Claims:**

I have checked their proofs for their theoretical claims, it is almost the same derivation steps from the strategic central limit theorem [1] with modification of including the weighted version.


**Ref:**

[1] Chen, Zengjing, Shui Feng, and Guodong Zhang. "Strategy-driven limit theorems associated bandit problems." arXiv preprint arXiv:2204.04442 (2022). https://arxiv.org/pdf/2204.04442

---

> ### Author Rebuttal · Authors · 2025-03-31
>
> We sincerely thank the reviewer for the comprehensive and insightful feedback. Below, we provide a point-by-point response to the issues raised:
>
> **1. The Use of “Exchangeability”**
>
> We appreciate the reviewer’s observation regarding the term “exchangeability” and fully agree that a clear definition is essential. In the revised manuscript, we have explicitly defined exchangeability here as referring to the rewards derived from the sequence of arm choices.
>
> **2. Presentation Adjustments**
>
> - **Two-Arm Bandit Introduction:**
>
> We acknowledge the suggestion regarding the introduction of the full term “Two-Armed Bandit” before using its acronym. In the revised version, we ensure clarity by introducing the full term at first mention (e.g., “Two-Armed Bandit (TAB)”).
>
> - **Typo Corrections:**
>
> Line 193: We have corrected the typo from “mean” to “main”.
>
> Line 410: We have changed the paragraph title from “Another simulations” to “More ML-based simulation studies” for clearer expression.
>
> - **Figure Improvements:**
>
> In Figure 2(b), we will add a detailed legend to describe each curve, ensuring that the visual representation is self-explanatory. The pink dashed, cyan solid, and orange dotted lines represent $\sigma=0.5, \sigma=0.6$, and $\sigma=1.0$, respectively. Additionally, we will update the caption of Figure 2(b) to “The empirical type I error rate across different $\lambda$ and $\sigma$, fixed $n=20000$”.
>
> - **Section Transitions and Introductions:**
>
> We agree that additional contextual transitions at the beginning of sections would improve readability. We will include brief overviews in Section 3.1 and elsewhere to better guide the reader through our arguments and experimental results.
>
> - **Lemma 2.1 Clarification:**
>
> We have revised Lemma 2.1 to explicitly define the alternative hypothesis $\mathcal{H}_1$ before applying it, thereby eliminating potential ambiguity.
>
> **3. Theoretical and Experimental Clarifications**
>
> Regarding the reviewer’s question about Lemma 2.1, its proof is provided in Chen et al.'s article (Z. Chen et al., “Strategy-driven limit theorems associated bandit problems,” Theorem 3.3). To improve the readability of our manuscript, we will include the full proof in the appendix of the revised version.
>
> **4. Ensemble Method (Stacking) Performance**
>
> We thank the reviewer for highlighting the performance discrepancies of the stacking method in the synthetic experiments. We identify two key factors contributing to this discrepancy:
>
> - First, the current implementation uses a limited selection of primary learners. We are actively investigating the incorporation of additional machine learning models as primary learners to enhance the performance of the stacking method.
>
> - Second, the choice of primary learners and their respective weights in the ensemble may not be optimal under all configurations, leading to suboptimal aggregation of predictions. We are exploring the use of more advanced meta-learners (e.g., random forests) instead of simple linear regression to better assign weights to different primary learners and further improve the stacking method’s performance.
>
> **5. Additional Evaluations on Real-World Data**
>
> We appreciate the reviewer’s suggestion regarding a more comprehensive evaluation on real-world data. To address this, we have conducted additional experiments using synthetic data based on real-world data. The results of these additional experiments are summarized in Table 1.
>
> **Table 1: Type I error rates and statistical power based on synthetic data derived from real-world dataset.**
>
> | Method   | Metric       | PWTAB | WTAB | $z$-DML | CUPED | DIM  |
> |----------|--------------|-------|------|---------|-------|------|
> | LightGBM | Type I Error | 0.052 | 0.052| 0.044   | 0.050 | 0.048|
> |          | Power        | 0.758 | 0.738| 0.744   | 0.740 | 0.498|
> | XGBoost  | Type I Error | 0.052 | 0.034| 0.046   | 0.050 | 0.048|
> |          | Power        | 0.758 | 0.738| 0.746   | 0.740 | 0.498|
> | Stacking | Type I Error | 0.052 | 0.052| 0.046   | 0.050 | 0.048|
> |          | Power        | 0.764 | 0.732| 0.746   | 0.740 | 0.498|
>
> These results provide compelling evidence of the effectiveness of our proposed PWTAB method in real-world scenarios. When the null hypothesis holds, all methods maintain Type I error rates close to 0.05, preserving the reliability of statistical inference in practical settings. Under the alternative hypothesis, the proposed method consistently outperforms competing methods in terms of statistical power. PWTAB achieves the highest statistical power when used with LightGBM or XGBoost, and its performance is further enhanced when combined with the ensemble learning algorithm Stacking.
>
> We sincerely appreciate the reviewer’s constructive comments, which have been invaluable in improving the clarity, rigor, and overall impact of our work.

---

> > ### Comment · Reviewer_QGjr · 2025-04-04
> >
> > I thank the authors throughout the response.
> >
> > 1.  **On the use of exchangeability**:
> >
> > Having a clearer explanation of exchangeability can significantly enhance both the readability and where the impact of the current work lies in. In your revision, please clearly highlight the advantages of your proposed approach compared to traditional A/B testing. Specifically, with the use of the strategy-driven limit theorem framework, the underlying test statistics no longer require the assumption of normality—an assumption typically made in traditional A/B testing. Therefore, your approach offers superior control over Type I errors and increased statistical power.
> >
> > 2. **On better presentation of your work**:
> >
> > As per Reviewer KBY3 mentioned, the authors should be mindful of the presentation. That is, the authors should either be less reliant on mathematical equations or clearly articulate the intuition behind each mathematical expression. Even in sections that do not need mathematical expression, the authors should still need to ensure that your work clearly convey intuition, provide smooth transitions between ideas. For example, line 241 to 257 could have been having better intuition and transition. That is, in line 243 to 249, the authors could instead say:
> >
> > " **Traditional methods such as CUPAC**, which rely solely on linear regression, might fail to capture these intricate patterns. **To overcome this limitation**, advanced machine learning methods are introduced. Specifically, LightGBM (Ke et al., 2017)—a state-of-the-art gradient boosting algorithm—is employed within the double machine learning (DML) framework (Chernozhukov et al., 2018).
> >
> > **Intuitively, the DML approach mitigates overfitting and reduces regularization biases by partitioning the dataset into multiple subsets. Each subset is used iteratively to estimate conditional relationships, ensuring robustness and improved predictive performance.** "
> >
> > There are additional sections where the presentation could be improved; however, I leave it to the authors to identify and enhance these sections on their own.
> >
> > 3. **On stacking methods**:
> >
> > Thank you so much for your clarification. It would be great to include them into your discussion section.
> >
> > 4. **On data-driven lambda**:
> >
> > I agree with Reviewer KBY3. It would be great to include some discussion on how to choose lambda in a data-driven approach.
> >
> > 5. **On the proof of Lemma 2.1**:
> >
> > Your theoretical results are primarily based on the work of Z. Chen et al. ("Strategy-driven limit theorems associated with bandit problems"). To ensure your manuscript is self-contained, please also include the detailed derivation of Lemma 2.1 in your appendix.
> >
> > 6. **On your additional Evaluations on Real-World Data**:
> >
> > I request that the authors clearly explain the rationale behind generating synthetic data from real-world data. Couldn't the experiments not be conducted directly using real-world data?
> >
> >
> > I thank the authors once again. I believe your manuscript is becoming clearer and, thus, better impact.

---

> > > ### Author Response · Authors · 2025-04-09
> > >
> > > We sincerely thank you for your continued positive feedback and insightful suggestions on our novel strategic A/B testing method. Through the revisions outlined below, we aim to further strengthen our manuscript and earn your full support.
> > >
> > > **On the use of exchangeability**
> > >
> > > We fully agree with your observation. Our proposed test statistic exhibits a concentrated, spike-like distribution around zero under the null hypothesis and a bimodal distribution away from zero under the alternative. By not relying on the normality assumption, our approach achieves superior control over Type I errors while enhancing statistical power. To highlight these advantages, we have revised the manuscript to explicitly compare our method with traditional A/B testing, emphasizing the flexibility afforded by the strategy-driven limit theorem framework.
> > >
> > > **On better presentation of our work**
> > >
> > > We value your guidance on improving readability. In addition to addressing the issue you pointed out regarding lines 241 to 257, we have minimized the use of extraneous mathematical formulas in the revised manuscript, retaining only those essential to our research. For these, we have added clear, intuitive explanations. Additionally, we have enhanced the logical flow and coherence throughout the text.
> > >  For example, the section from line 271 (left) to line 220 (right) has been revised as follows:
> > >
> > > “To address this issue, we perform multiple samples reorderings, repeatedly calculate the $p$-value of $T_{n,\lambda}(\theta_n^*)$, and aggregate these via meta-analysis to enhance the robustness of statistical inference.”
> > >
> > > The section from line 232 (right) to line 237 (right) has been revised to:
> > >
> > > “However, varying sample orderings can yield inconsistent $p_{\lambda}^{(b)}$ values, and the conclusions drawn from individual $p$-values may be unclear. To resolve this, we apply meta-analysis to synthesize an overall $p$-value, improving the reliability of the results derived from individual $p_{\lambda}^{(b)}$ values (Walker et al., 2008; Lee, 2019).”
> > >
> > > **On data-driven $\lambda$**
> > >
> > > We fully agree with you and Reviewer KBY3 on the importance of a data-driven approach to selecting $\lambda$. As detailed in our rebuttal to Reviewer KBY3, we have proposed a data-driven approach for selecting $\lambda$, which we have now incorporated into the revised manuscript for clarity and completeness.
> > >
> > > **On the proof of Lemma 2.1**
> > >
> > > We fully agree with you. To ensure the coherence of the paper, we have independently included the detailed derivation of Lemma 2.1 in the appendix of the latest revised version.
> > >
> > > **On the additional Evaluations on Real-World Data**
> > >
> > > We appreciate the opportunity to clarify the rationale behind this approach. Our decision to generate synthetic data stems from two key practical constraints associated with real-world A/B testing datasets:
> > >
> > > - Limited Availability of Real-World Data: Real-world A/B testing datasets are often constrained in size and scope, which can limit their suitability for comprehensive statistical evaluations. Synthetic data allows us to scale experiments and explore a wider range of scenarios while preserving the distributional characteristics of real-world data.
> > >
> > > - Absence of Ground Truth for Strategy Improvements: The average treatment effect in real-world datasets is typically unknown, making it difficult to accurately estimate critical metrics such as empirical Type I error rates and statistical power—both essential for validating our method’s performance. By generating synthetic data based on real-world data, we can control the average treatment effect while preserving the original data distribution, thereby enabling precise and reliable estimation of these metrics.
> > >
> > > We hope these clarifications and revisions fully address your concerns. Thank you again for your valuable input, which has greatly improved our manuscript.

---

### Official Review · Reviewer_KBY3 · 2025-03-20

**Overall Recommendation:** 3

**Summary:**

This paper introduces a novel approach to A/B testing focused on detecting minor average treatment effects (ATEs) in large-scale applications. The authors propose a maximum probability-driven two-armed bandit process with a weighted mean volatility statistic and incorporation of permutation methods. The key theoretical contribution is the strategic central limit theorem (SCLT), which yields more concentrated distributions under the null hypothesis and less concentrated distributions under alternatives, thereby enhancing statistical power.

The proposed permuted weighted two-armed bandit (PWTAB) method incorporates doubly robust estimation for counterfactual outcomes. Experiments on both synthetic and real-world ride-sharing company data demonstrate PWTAB consistently outperforms standard methods like DIM, CUPED, and z-DML while maintaining proper Type I error control.

**Claims And Evidence:**

The claims are generally well-supported by evidence:

- The central claim that WTAB improves statistical power is backed by both theoretical analysis (SCLT) and empirical results showing superior performance in different simulation settings.
- Type I error control is verified through comprehensive simulation studies in Table 2, with empirical rates remaining close to the nominal α=0.05 level across varied configurations.
- Empirical evidence in Figure 4 demonstrates PWTAB consistently outperforms comparison methods, particularly for nonlinear functions.

**Essential References Not Discussed:**

Nothing completely relevant seems to be omitted from the manuscript

**Experimental Designs Or Analyses:**

The experiments are thorough and well-designed:

- Synthetic data tests span 32 configurations combining four different functions F(X1,X2), four G(X1,X2) (including two null hypotheses GI, GII), and two noise levels (σε=0.5, 0.6).
- Sample size n=20,000 realistically represents large-scale A/B testing scenarios.
- Real-world validation uses three datasets (A, B, C) from a ride-sharing company, with results in Figure 5 showing PWTAB achieves smaller p-values regardless of the machine learning algorithm used.

The authors rigorously compared their approach against DIM, CUPED, and z-DML baselines, showing consistent improvements particularly for nonlinear function settings.

**Methods And Evaluation Criteria:**

The methodological approach effectively addresses the problem of detecting minor treatment effects:

- The weighted mean-volatility statistic (Eq. 5) provides a flexible framework balancing detection power with Type I error, with weight parameter λ carefully chosen to maximize statistical power.
- The permutation-based approach (Algorithm 1) using Cauchy combination addresses the "p-value lottery" problem, with B=25 permutations determined sufficient through empirical testing.

The evaluation criteria include both Type I error control and statistical power across varied conditions (linear/nonlinear functions, heterogeneous effects, different noise levels σε ∈ {0.5, 0.6}).

**Other Comments Or Suggestions:**

Nothing to add here.

**Other Strengths And Weaknesses:**

**Strengths:**

- Addresses an economically significant problem with a theoretically grounded solution.
- The integration of bandit algorithms with traditional A/B testing creates an innovative hybrid methodology.
- Demonstrates superior performance for nonlinear relationships where CUPED falters.
- The doubly robust estimation approach provides protection against model misspecification.

**Weaknesses:**

- The mathematical density may limit adoption by practitioners without strong statistical backgrounds.
- Limited guidance on practical λ selection beyond the 0.03 threshold.
- The paper could better explain the intuition behind why breaking exchangeability improves performance.

**Questions For Authors:**

1. Beyond the threshold approach, are there data-driven methods to select optimal $\lambda$ values?
2. How well does the method generalize to domains beyond ride-sharing (e.g., e-commerce) where metrics and effect sizes differ?

**Relation To Broader Scientific Literature:**

The paper effectively connects to relevant literature across:

- A/B testing
- Causal inference
- Multi-armed bandits
- Permutation tests

**Theoretical Claims:**

The theoretical proofs seems sound and rigorous, building on:

- Theorem 4.1 establishes that the asymptotic distribution follows a spike distribution.
- Theorem 4.2 demonstrates Type I error control and consistency against fixed alternatives, under $H_0$.
- The weighted statistic maintains the same optimal policy structure (Eq. 10), with $λ$ constrained by threshold $f(\lambda)≤ 0.03$ to ensure proper convergence.

---

> ### Author Rebuttal · Authors · 2025-03-31
>
> We sincerely appreciate the reviewer’s careful evaluation of our work and the constructive feedback provided. Below, we address the key weaknesses and questions raised:
>
> **1. Adding a More Intuitive Explanation of Mathematical Densities**
>
> We thank the reviewer for this valuable suggestion. We acknowledge that the extensive mathematical derivations may pose a barrier to practitioners with limited statistical backgrounds. In the revised version, we have incorporated an intuitive explanation section to explain the form of different probability densities under different hypotheses. We believe this additional exposition will facilitate a broader understanding and practical application of our method.
>
> To illustrate, consider the case when the null hypothesis holds with $\lambda$ fixed. Given that the optimal policy parameter $\vartheta_1^*$ has an equal probability of being 0 or 1, assume that $R_1^{(1)}$ is observed and that $T\_{1, \lambda}(\theta_1^*)\ge 0$. Consequently, according to the optimal policy, $\vartheta_2^*=1$, implying that $R_2^{(1)}$ will be observed. This process continues with $\vartheta_i^*=1$ until there exists some index $m$ such that $T\_{m, \lambda}(\theta_m^*)<0$. Under the assumption that the null hypothesis holds, it is likely that $T\_{2, \lambda}(\theta_2^*)<0$, resulting in $\vartheta_3^*=0$, which leads to the observation of $R_3^{(0)}$, a reward that is more likely to exceed 0. This brief discussion shows that the optimal policy $\theta_n^*$ will control the value of $T\_{n, \lambda}(\theta_n^*)$ to fluctuate around 0 under the null hypothesis, thereby concentrating its distribution around 0. A similar rationale applies when the alternative hypothesis holds.
>
> **2. Guidance on $\lambda$ Selection**
>
> We agree that the guidance on selecting $\lambda$ is crucial. The threshold value of 0.03 was derived empirically from our synthetic experiments. However, we are actively exploring more data-driven methods for selecting $\lambda$. We propose a data-driven approach for selecting $\lambda$ by first discretizing its range and then employing bootstrapping techniques to generate multiple datasets. For each candidate $\lambda$, we compute the type I error rate across these datasets. The optimal $\lambda$ is chosen as the one that maximizes statistical power while controlling the type I error.
>
> **3. More Real-world Applications**
>
> We appreciate the reviewer’s insightful query on the generalizability of our method to domains such as e-commerce. Although our current real-world validation is based on ride-sharing data, our preliminary experiments in other domains indicate that the method demonstrates strong potential. We are confident that our approach can be generalized to most companies conducting A/B testing. We intend to extend our experimental evaluation to include additional domains, such as a food delivery company and an internet technology company, thereby providing a more robust demonstration of the method’s versatility and robustness.
>
> **4. Breaking Exchangeability**
>
> We appreciate the reviewer’s interest in this aspect. We will clarify the advantages of breaking exchangeability in the revised manuscript by explicitly detailing how it contributes to enhanced performance. Traditional hypothesis testing methods based on the Central Limit Theorem (CLT) are inherently data-driven; once i.i.d. samples are observed, the construction of the test statistic is independent of the sample order, implying that the data are exchangeable. In contrast, our proposed testing framework is goal-driven—it seeks to progressively construct the test statistic from the available data to maximize statistical power. In our proposed two-armed bandit framework, earlier data actively influences the construction of the current test statistic, making the data  non-exchangeable. This shift toward a maximum-probability objective enables the optimal construction of the test statistic, thereby enhancing testing performance.
>
> Once again, we are grateful for the reviewer’s positive comments and valuable suggestions. We are committed to incorporating these improvements to enhance the clarity, interpretability, and impact of our work.

---

### Decision · Program_Chairs · 2025-05-01

**Decision:**

Accept (poster)

**Comment:**

This paper received two positive scores (Accept and Weak Accept) and one Reject. Upon initiating discussion between the reviewers, it was found that the reviewer with negative score had a main concern about the correctness of one of the steps in the theorem proofs. However, from the reviewer discussion, reviewer QGJr (positive review) chimed in to clarify that the concern is mitigated and they believe the proof provided is sound.

In light of this, it seems to me that the main concern of the reviewer voting “Reject” should be dispelled, hence recommending to accept this paper.